# Multi-site fungicides suppress banana Panama disease, caused by *Fusarium oxysporum* f. sp. *cubense* Tropical Race 4

**Stuart Cannon[1,2‡], William Kay[1,3‡], Sreedhar Kilaru[1‡], Martin Schuster[1‡], Sarah Jane Gurr** [1,4]*, **Gero Steinberg** [1,4]*

**1** Biosciences, University of Exeter, Exeter, United Kingdom, **2** Institute of Biomedical and Clinical Science, University of Exeter, Exeter, United Kingdom, **3** Department of Plant Sciences, University of Oxford, Oxford, United Kingdom, **4** University of Utrecht, Utrecht, The Netherlands

‡ These authors contributed equally and are listed alphabetically
* s.j.gurr@exeter.ac.uk (SJG); g.steinberg@exeter.ac.uk (GS)

## Abstract

Global banana production is currently challenged by Panama disease, caused by *Fusarium oxysporum* f.sp. *cubense* Tropical Race 4 (FocTR4). There are no effective fungicide-based strategies to control this soil-borne pathogen. This could be due to insensitivity of the pathogen to fungicides and/or soil application *per se*. Here, we test the effect of 12 single-site and 9 multi-site fungicides against FocTR4 and Foc Race1 (FocR1) in quantitative colony growth, and cell survival assays in purified FocTR4 macroconidia, microconidia and chlamydospores. We demonstrate that these FocTR4 morphotypes all cause Panama disease in bananas. These experiments reveal innate resistance of FocTR4 to all single-site fungicides, with neither azoles, nor succinate dehydrogenase inhibitors (SDHIs), strobilurins or benzimidazoles killing these spore forms. We show in fungicide-treated hyphae that this innate resistance occurs in a subpopulation of "persister" cells and is not genetically inherited. FocTR4 persisters respond to 3 µg ml$^{-1}$ azoles or 1000 µg ml$^{-1}$ strobilurins or SDHIs by strong up-regulation of genes encoding target enzymes (up to 660-fold), genes for putative efflux pumps and transporters (up to 230-fold) and xenobiotic detoxification enzymes (up to 200-fold). Comparison of gene expression in FocTR4 and *Zymoseptoria tritici*, grown under identical conditions, reveals that this response is only observed in FocTR4. In contrast, FocTR4 shows little innate resistance to most multi-site fungicides. However, quantitative virulence assays, in soil-grown bananas, reveals that only captan (20 µg ml$^{-1}$) and all lipophilic cations (200 µg ml$^{-1}$) suppress Panama disease effectively. These fungicides could help protect bananas from future yield losses by FocTR4.

## Author summary

Bananas are amongst the most popular fruits eaten world-wide, yet their production is seriously challenged by the fungus *Fusarium oxysporum* f. sp. *cubense*, Tropical Race 4 (FocTR4). Hitherto, no effective strategy to control this devastating disease has been

**Data Availability Statement:** The authors confirm that all relevant data are included in the paper or in the Supporting Information file. Additional information is available from the authors upon

request. Raw sequencing data are available from the NCBI Sequence Read Archive (https://www. ncbi.nlm.nih.gov/bioproject/PRJNA803733). Numerical data and statistical analysis are provided in the Supplementary Information file S1 Data.

**Funding:** This work was supported by the Biotechnology and Biological Sciences Research Council (https://www.ukri.org/councils/bbsrc/; grants: BB/N020847/1, led by Dr. Dan Bebber with SJG and GS, and in association Prof. Gert Kema, Wageningen, The Netherlands; BB/P018335/1, led by GS with SJG). The funders had no role in study design, data collection and analysis, decision to publish, or preparation of the manuscript.

**Competing interests:** We have read the journal's policy and the authors of this manuscript have the following competing interests: Aspects of the research described herein are covered by patent application GB2202216.4 (Fusarium treatment in soil with MALCs) and patent WO2020201698A1 (Antifungal Compositions).

described. Indeed, even fungicides, which are generally considered to be our "front-line weapon" against plant pathogenic fungi, are deemed ineffective against FocTR4. Here, we analyse the use of 12 single-site and 9 multiple-site fungicides against FocTR4 and FocR1 (*Fusarium oxysporum* f. sp. *cubense*, Race 1) and compare these findings with data raised from the wheat pathogen *Zymoseptoria tritici*. We perform quantitative growth and cell survival assays, using FocTR4 hyphae and the 3 spore types (macroconidia, microconidia, chlamydospores). We show that all FocTR4 morphotypes are highly tolerant of the most widely-used single-site fungicides (azoles, succinate dehydrogenase inhibitors, strobilurins). Analysis of gene expression in surviving "persister" hyphae suggests that they cope with single-target fungicides by multiple mechanisms. This includes increased production of fungicide target enzymes, efflux proteins and detoxification enzymes. Only multi-site fungicides, which have multiple ways to affect the pathogen cell, proved to be effective in killing FocTR4. However, quantitative assessment of disease symptom development in fungicide-treated bananas revealed that only captan and three lipophilic cations have the potential to control Panama disease.

## Introduction

Bananas are amongst the most popular fruits eaten world-wide. Indeed, bananas (including plantains) are the 4th most important global staple food crop, produced in over 150 countries at >114 million metric tons *per annum* (2019, http://www.fao.org/economic/est/est-commodities/oil crops/bananas). Bananas provide staple local food for almost half a billion people, whilst exports support economic stability [1]. They are thus an essential calorie crop and a revenue-generating trade commodity [2].

The world's banana supply was previously challenged by *Fusarium oxysporum* f. sp. *cubense* (Foc) as, in the 1950s, Race 1 (FocR1) decimated the commercially-dominant Gros Michel variety [1,3]. This threat was overcome by introduction of a resistant Cavendish variety [1]. However, with the appearance of Tropical Race 4 (FocTR4), identified in Taiwan in 1967, the world's banana supply faced renewed jeopardy [2]. This aggressive strain has spread across the continents, reaching South America in 2019 [4]. The threat of Panama disease is of high significance, as Cavendish bananas currently account for ~40% of world production and >90% of all exports [5].

FocTR4 (also named *Fusarium odoratissimum* [6]) produces 3 asexual spore forms, macroconidia, microconidia and chlamydospores [6]. Studies in Foc sub-tropical Race 4 suggest that these spore forms participate in the infection of bananas [7]. The thick-walled, melanised chlamydospores tolerate adverse environmental conditions and persist in soil debris for decades [3]. They spread as silent passengers on non-host species or are carried in soil, on footwear or on farm equipment [1]. Chlamydospores germinate under favourable conditions [8], penetrate root tissues, invade the cortex and enter the endodermis, where they colonise and block xylem vessels, inciting host tyloses formation [3]. The conidial forms are thought to play important roles during colonization [6]. Infection results in wilted, yellowed leaves, vasculature browning, corm necrosis and plant death [1].

Different strategies to control FocTR4 include (i) crop husbandry, including removing infected tissues [9], (ii) biological control agents [10], (iii) efforts to generate resistant varieties [5,11] (iv) biosecurity protocols to reduce spread [10]. However, none of these have been particularly effective [1]. The best way to control fungi on crops is with fungicides [12]. Moreover, disinfectants are used to clean contaminated agricultural equipment [13,14,15,16]. Studies on

the effect of various fungicides on FocTR4 *in vitro* [13,17,18] or *in vivo*, by corm injection [19], bare root immersion and/or soil drenches [13,17,18,20] provided contradictory results. Consequently, it is thought that "chemical measures are of limited or questionable efficacy" [1,9], or "*Fusarium* wilt cannot be controlled by fungicides" (ProMusa, https://www.promusa.org/ Fungicides +used+in+banana+plantations). It is currently not known whether the insensitivity of *Fusarium* is related to the application of fungicides to soil or to the physiological response of the pathogen itself. Indeed, most highly effective and widely used fungicides, such as azoles, succinate dehydrogenases inhibitors (SDHIs) and strobilurins, target a single enzyme catalysing an essential process. These single-site fungicides can be overcome by point mutations in their respective target gene or by increased expression of mutated isoforms of the target enzyme [21,22]. Such increased transcription can be a consequence of mutations in transcription factors and thus is genetically inherited to the next generation of pathogen cells [23,24]. This risk of resistance development can be minimised by using multi-site inhibitors, which have multiple modes of action, thereby avoiding target-site resistance-related mechanisms [25].

Here, we address these claims and develop robust and quantitative assays to evaluate 21 fungicides from 7 classes on (i) *in vitro* FocTR4 and FocR1 colony formation and, for comparison, the fungicide-sensitive wheat pathogen *Zymoseptoria tritici*, (ii) survival of FocTR4 macrospores, microspores, chlamydospores in liquid culture. This gave a subset of fungicides to test for their potential to protect against Panama disease caused by these pathogens. We then focussed our study on the current threat to Cavendish banana production by FocTR4, and compared the transcriptional responses of FocTR4 and *Z. tritici* to the single-site targeting azoles, SDHIs and strobilurins. This analysis provided better insight into the molecular drivers of tolerance against these fungicide classes. This systematic approach enabled us to (i) describe likely reasons for the failure of major fungicide classes to control FocTR4 (ii) identify that captan and mono-alkyl lipophilic cations (MALC$_S$) suppress Panama disease.

## Results

### All morphotypes of FocTR4 cause Panama disease in bananas

Plant pathogenic fungi colonise their hosts by invasive hyphal growth. To visualise FocTR4 banana root colonisation, we made a fluorescently-tagged version of the FocTR4 plasma membrane-located Sso1-like syntaxin homologue (FocSso1, UniProt ID: X0JJC7, similar to *S. cerevisiae* Sso1p; BLAST P = 3e-31) previously used to visualise *Z. tritici* in host tissues [26]. Expression of the GFP-FocSso1 fusion protein resulted in green plasma membrane in FocTR4 hyphae *in vitro* (Fig 1A). However, as chlamydospores are most relevant for *Fusarium oxysporum* in field infections [3], we developed a protocol to purify GFP-FocSso1-expressing chlamydospores (see Materials and Methods) and inoculated plants by drenching their roots with these spores. We found chlamydospores germinating on the root surface by 7 days post inoculation (dpi, Fig 1B). This transition is likely "fuelled" by endogenous resources. Indeed, ultrastructural studies and live cell imaging of BODIPY 493/503-stained chlamydospores revealed prominent lipid droplets (Fig 1C and inset, lipid droplets indicated by "LD"). At later stages of infection, invasive hyphae colonise the root, breaching plant cell walls and spread within host cells (Fig 1D, 26 dpi).

Besides hyphae and chlamydospores, FocTR4 forms 2 other known asexual spore types, namely microconidia and macroconidia [6]. We asked whether all morphotypes cause Panama disease. To this end, we developed protocols to purify the 3 spore forms and hyphae. Morphological assessments revealed the preparations are ~94–100% pure (Fig 1E). We confirmed this by preparing RNA from all morphotypes to determine their respective transcriptomes. We

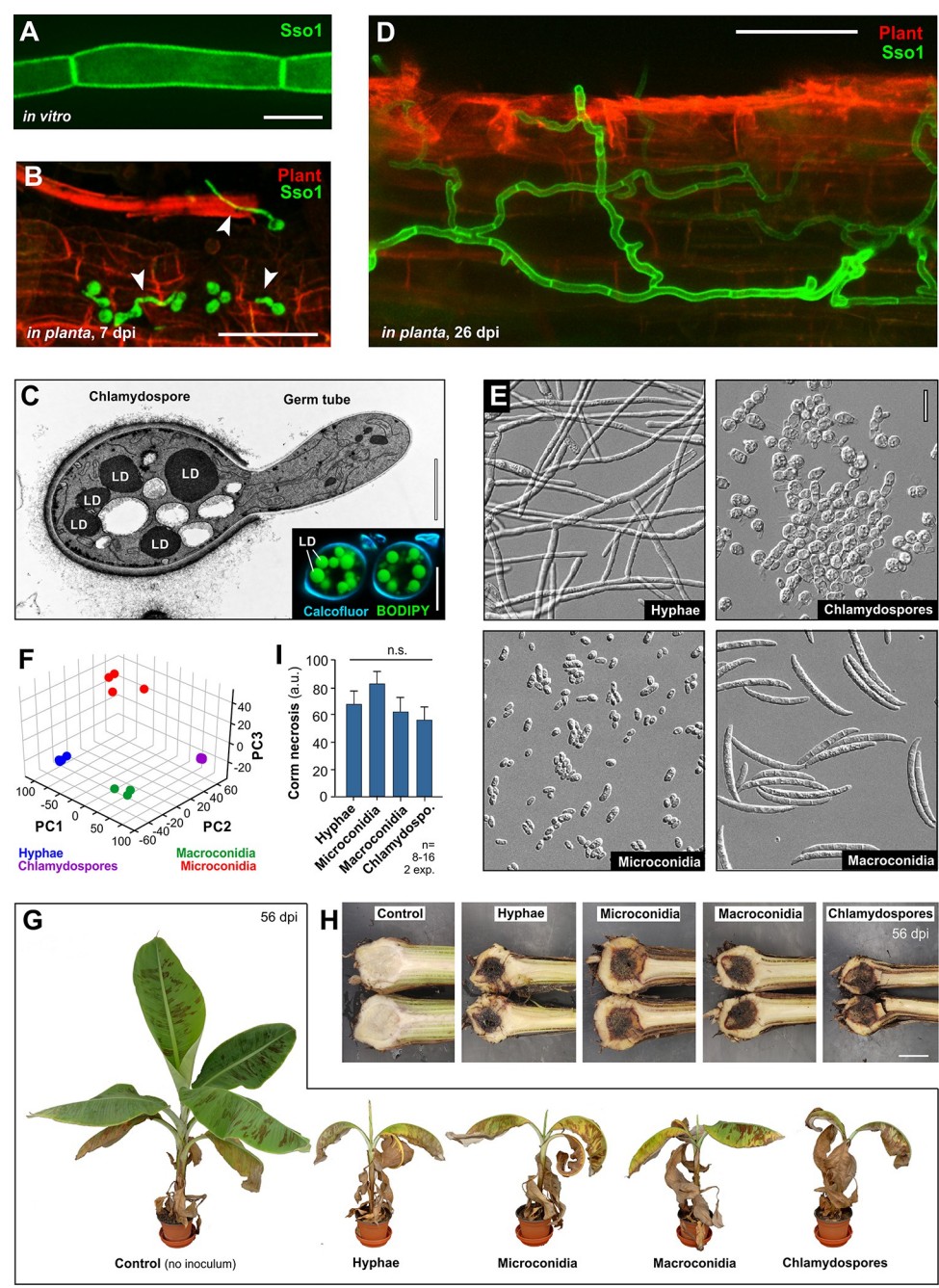

**Fig 1. Pathogenicity and transcriptome of FocTR4 morphotypes. A** Hyphal cells of FocTR4, expressing the plasma membrane marker Foc_GFP-Sso1 grown in liquid medium. Scale bar = 5 μm. **B** Chlamydospores, expressing the plasma membrane marker Foc_GFP-Sso1 (green) on the surface of a root (red) at 7 dpi. Plant cell walls are visualised by their auto-fluorescence. Germ tubes are indicated by arrowheads. Scale bar = 50 μm. **C** Lipid droplets (LD) in electron micrograph of a germinating chlamydospores and in a chlamydospores, stained with BODIPY 493/503 (green in inset; cell wall stained using calcofluor). Germination was triggered by incubation in PDB for 5 h. Scale bars = 2 μm and 5 μm (inset). **D** FocTR4 hyphae colonising banana root tissue at 26 dpi. Fungal cells express a fluorescent plasma membrane-located syntaxin (eGFP-Sso1, green), plant cell walls are visualised by their auto-fluorescence (red). Scale bar = 50 μm. **E** Morphology of purified FocTR4 hyphae, microconidia, macroconidia and chlamydospores. Scale bar = 20 μm. **F** Principal component analysis of RNA-preparations from 4 morphotypes of FocTR4. See also S1 Video. **G** Banana plants at 56 days after inoculation (dpi) with hyphae, microconidia, macroconidia or chlamydospores. All morphotypes cause leaf wilting, chlorosis and necrosis. **H** Necrosis of banana corms at 56 dpi with hyphae, microconidia, macroconidia or chlamydospores. Scale bar = 3 cm. **I** Quantitative assessment of necrosis in corm tissue after inoculation with the morphotypes. Bars in (**I**) represent mean ± SEM; sample sizes are indicated; statistical

analysis in (**I**) used one-way ANOVA; n.s. = non-significant difference amongst data sets to control at two-tailed
P = 0.2590.

analysed 4 mRNA preparations for each morphotype and compared their similarity by princi-pal component analysis [27], using DESeq2 v1.14.1 (URL and references to bioinformatic tools in S1 Table). The resulting plot showed clusters of sample "dots" for each morphotype that were clearly separated from each other (Fig 1F and S1 Video). Thus, both morphological and transcriptional profiling supports the notion that each of our preparations comprise a separate morphotype of FocTR4.

We designed a quantitative whole plant-in-soil infection assay to investigate if all morpho-types cause disease. This used pre-weighed soil and defined amounts of spore suspension, sup-plemented with the non-ionic adjuvant surfactant Silwet L-77 (to ensure even distribution in the inoculated soil; see Materials and Methods). 8–12 week-old bananas, growing in soil, were inoculated with hyphae, microconidia, macroconidia or chlamydospores, and assessed disease symptoms after 56 days grown at $27 \pm 1°C$. This reflects the optimum temperature for banana productivity [28]. All morphotypes cause wilting, with necrotic and/or chlorotic leaves (Fig 1G). We quantified corm necrosis, using image analysis techniques (Fig 1H, see details in Materials and Methods). This disease symptom is widely accepted as an indicator of FocTR4 infection [6,29]. No significant difference between fungal morphotypes was found (Fig 1I; one-way ANOVA test, not different at two-tailed P = 0.2590). Thus, we conclude that all FocTR4 morphotypes can cause Panama disease.

## FocTR4 and FocR1 form colonies in the presence of major fungicide classes

Panama disease of bananas is considered uncontrollable by fungicides [1,9]. This may either be due to resistance of the pathogen and/or a hindering effect of soil due to its interaction with fungicides [30,31,32]. To test the hypothesis that FocTR4 and FocR1 show innate fungicide resistance, a quantitatively assessment of microconidia growth on fungicide-supplemented potato dextrose agar (PDA) plates, using 21 compounds from 7 main fungicide classes, was performed. The FocTR4 and FocR1 colonies grew as a peripheral "corona" of hyphae, with central "yeast-like" cells (Fig 2A–2C, FocTR4; Fig 2D–2F, FocR1). We noticed that most fungi-cides inhibited peripheral hyphal growth before affecting the central "yeast-like" cells (S1A Fig, FocTR4 on dodine-supplemented plates shown as example). This suggests that the colony diameter, as described [13,17,18], is an inaccurate measure of fungicide efficacy in FocTR4 and FocR1. We thus developed a different method, using the bright appearance of colonies on dark agar plates, in digital images, as a measure of growth (see Methods). This provided dose-response curves for both fungi, exposed to all fungicides (Fig 3). We determined fungicide concentrations leading to 50% growth inhibition ($EC_{50}$), 90% growth inhibition ($EC_{90}$) and >99.5% growth inhibition (minimal inhibitory concentration = MIC). FocTR4 and FocR1 show similar responses to fungicides (Table 1 and Fig 3), with slight variations between both races. FocR1 and FocTR4 were prevented efficiently from plate growth by carbendazim (MIC<1 µg ml$^{-1}$) and 3 other compounds were effective at MIC<5 µg ml$^{-1}$ (Table 1). Azoles showed moderate activity in FocR1 and FocTR4 (MIC~8–67 µg ml$^{-1}$), whereas 12 fungicides had low inhibitory effects on FocTR4 and FocR1 (MIC>100 µg ml$^{-1}$, Table 1). Amongst these were 8 fungicides that did not fully inhibit colony formation (MIC>1000 µg ml$^{-1}$; Table 1, Fig 3, S1B Fig, which shows FocTR4 on Boscalid-supplemented plates shown as an example). To exclude that the insensitivity of FocTR4 and FocR1 are due to our experimental conditions, plate growth of the fungicide-susceptible wheat pathogen *Zymoseptoria tritici* (wildtype strain IPO323) under comparable conditions was assessed. We found that IPO323 was highly

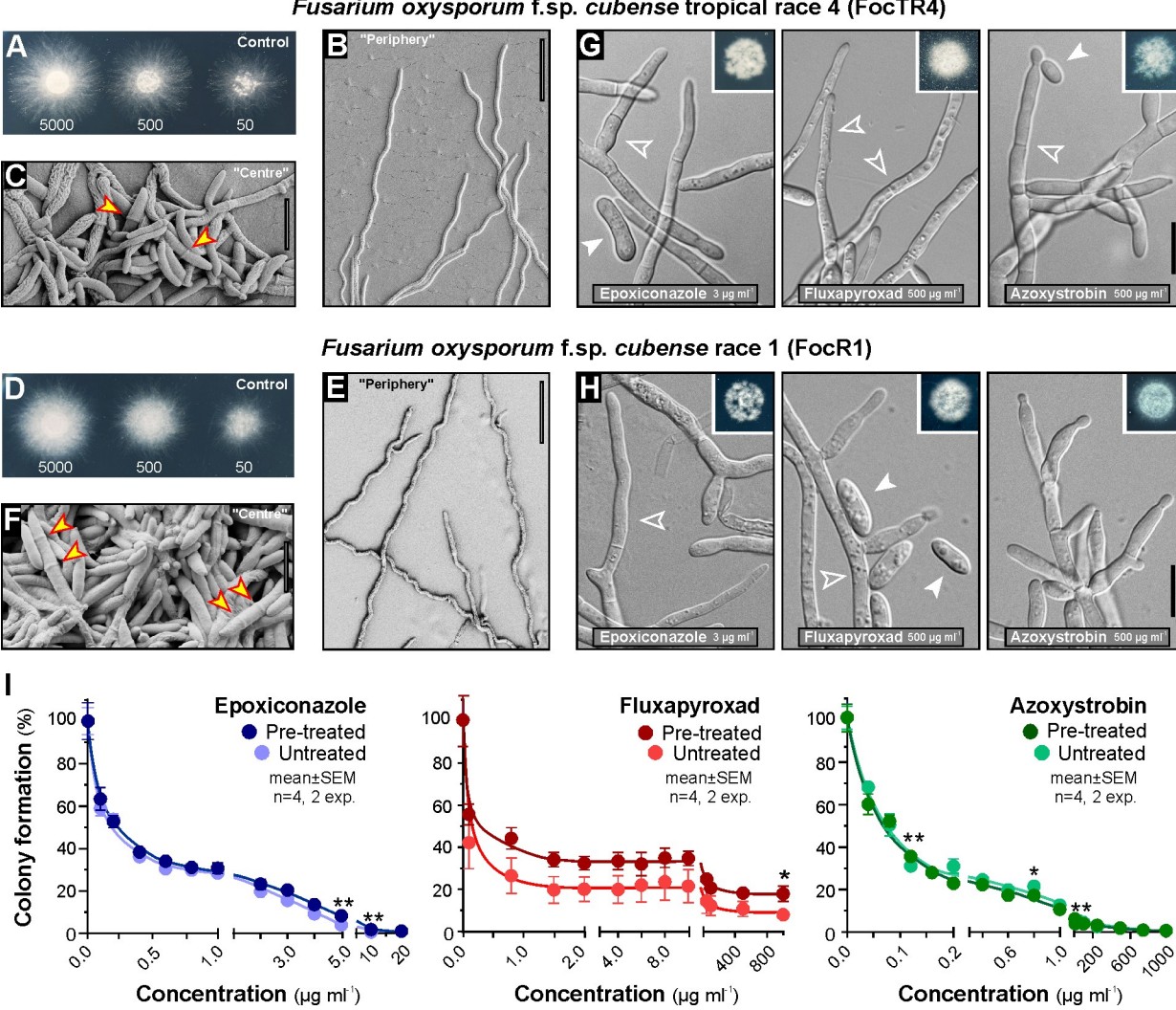

**Fig 2. The effect of various fungicides on growth of FocTR4 and FocR1 on agar. A** FocTR4 colonies on PDA plates after 2 days growth at 25°C. Note that colonies contain of central region of densely grown cells and a "corona" of more diffuse spreading hyphae. Numbers of inoculated microconidia per colony are indicated. **B**, **C** Scanning electron micrographs of cells at the edge (**B**) and the centre (**C**) of a colony of FocTR4, grown on PDA agar plates. The central region shows yeast-like cells, which are occasionally multi-cellular (septa indicated by arrowheads), thus may resemble micro- or macroconidia; the peripheral region consists of hyphae. Scale bars = 10 μm (**C**) and 50 μm (**B**). **D** FocR1 colonies on PDA plates after 2 days growth at 25°C. Numbers of inoculated microconidia per colony are indicated. **E**, **F** Scanning electron micrographs of cells at the edge (**E**) and the centre (**F**) of a colony of FocR1, grown on PD agar plates. The central region shows yeast-like cells, which are occasionally multi-cellular (septa indicated by arrowheads), thus may resemble micro- or macroconidia; the peripheral region consists of hyphae. Scale bars = 10 μm (**F**) and 50 μm (**E**). **G**, **H** Morphology of FocTR4 (**G**) and FocR1 (**H**) persister cells, harvested from PDA plates after 2 days grown in the presence of 3 μg ml$^{-1}$ epoxiconazole, 500 μg ml$^{-1}$ fluxapyroxad or 500 μg ml$^{-1}$ azoxystrobin. Colonies from where cells were taken are shown in the upper right. Scale bars = 10 μm. **I** Colony growth of FocTR4 persisters on fungicide-supplemented agar plates. Pre-treated: Cells were pre-grown for 2d in liquid cultures, supplemented with 3 μg ml$^{-1}$ epoxiconazole, 500 μg ml$^{-1}$ fluxapyroxad or 500 μg ml$^{-1}$ azoxystrobin, harvested and plated on fungicide-containing agar plates; Untreated: Cells were pre-grown for 2d in liquid cultures, supplemented with 0.06–2.5% (v v$^{-1}$) of the solvent methanol, harvested and plated on fungicide-containing agar plates. Note that colony formation on fungicide-supplemented agar plates by liquid-grown persister cells is almost identical to that of control cells. Non-linear regressions in was done using GraphPad Prism 6 (equation "[Inhibitor] vs. response—variable slope (four parameters)"). All data points are given as mean ± SEM, sample size is 4 colonies from 2 experiments; statistical testing used Student's t-test, only significant differences are indicated (* represents two-tailed P-value <0.05; ** represents two-tailed P-value <0.01).

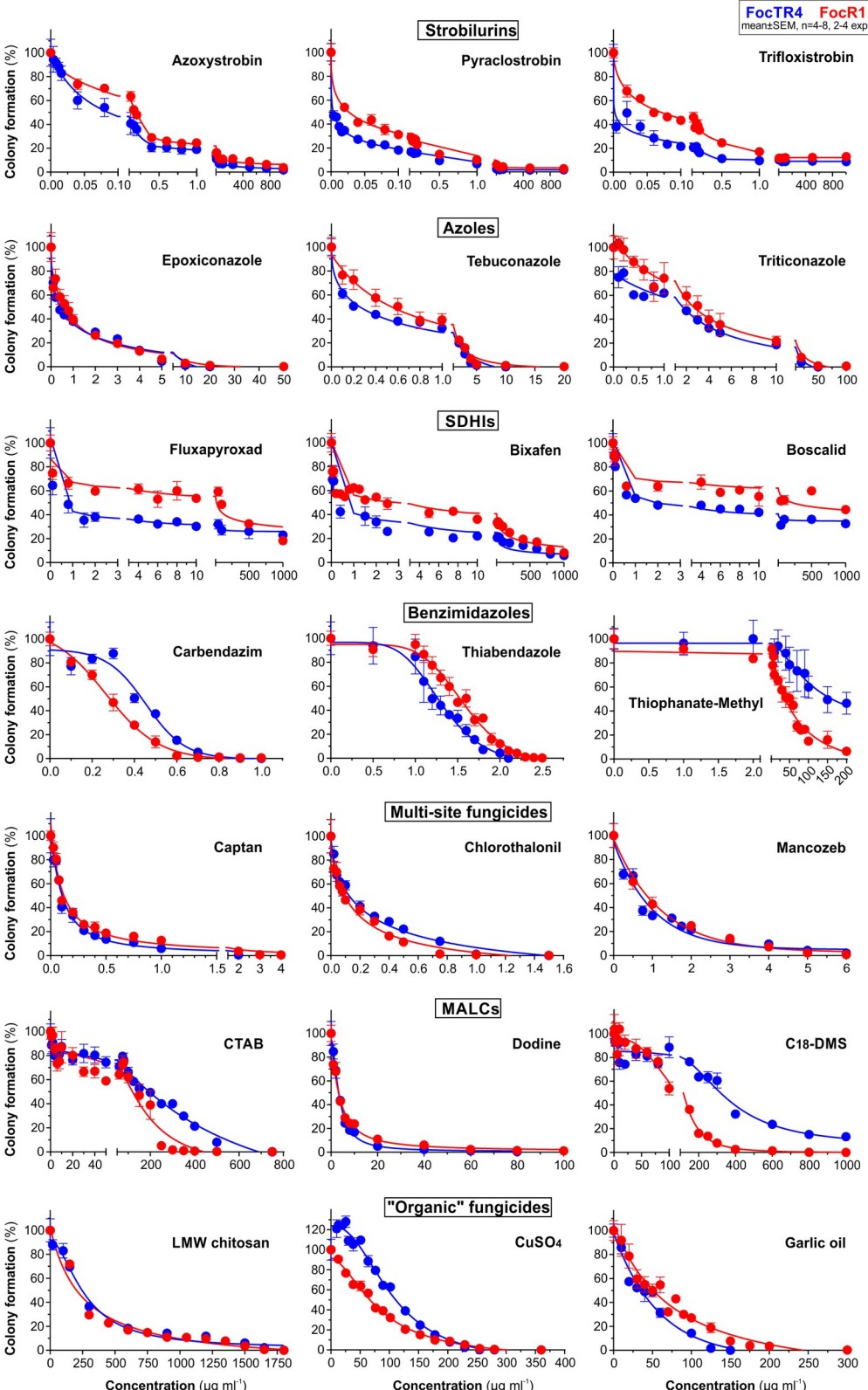

**Fig 3. Quantitative assessment of colony formation of FocR1 and FocTR4 on PDA plates, supplemented with fungicides.** Growth was monitored by measuring cell formation in images (see Methods) after 2d incubation at 25°C. 7 fungicide class were tested, each represented by 3 compounds; SDHIs = succinate dehydrogenase inhibitors;

MALCs = Mono-alkyl cations. Note that a small population of "persister cells" of FocTR4 and FocR1 are highly tolerant in the major fungicide classes (azoles, SDHIs and strobilurins). See also S1 Fig for inhibition curves of *Z. tritici* cells and Table 1 for estimated concentrations at 50%, 90% and >99.5% growth inhibition (EC$_{50}$, EC$_{90}$, MIC). Non-linear regressions analysis used GraphPad Prism 9.

sensitive to most fungicides (16 compounds with MIC<5 µg ml$^{-1}$, with 12 of these with MIC<1 µg ml$^{-1}$; 4 compounds with MIC<5 µg ml$^{-1}$; S2 Table and S1C–S1J Fig).

## FocTR4 forms fungicide-tolerant "persister" hyphae

In the presence of several azoles, strobilurins and SDHIs, colony growth of FocTR4 and FocR1 was strongly inhibited at low concentrations. Paradoxically, sparse growth occurred at higher

**Table 1. Inhibition of plate growth of FocTR4 and FocR1.**

| Fungicide | EC$_{50}$ [a,b] | | EC$_{90}$ [a,b] | | MIC [a,c] | |
|---|---|---|---|---|---|---|
| | FocTR4 | FocR1 | FocTR4 | FocR1 | FocTR4 | FocR1 |
| *Strobilurins* | | | | | | |
| Azoxystrobin | 0.083 | 0.180 | 14.340 | 76.920 | >1000[e] | >1000[e] |
| Pyraclostrobin | 0.005 | 0.028 | 0.801 | 1.930 | >1000[e] | >1000[e] |
| Trifloxystrobin | 0.020 | 0.073 | 9.050 | >1000[e] | >1000[e] | >1000[e] |
| *Azoles* | | | | | | |
| Epoxiconazole | 0.405 | 0.586 | 5.879 | 5.620 | 14.225 | 29.020 |
| Tebuconazole | 0.261 | 0.545 | 3.525 | 4.000 | 8.049 | 14.750 |
| Triticonazole | 1.699 | 2.675 | 15.520 | 22.480 | 40.625 | 66.750 |
| *Succinate dehydrogenase inhibitors* | | | | | | |
| Fluxapyroxad | 0.881 | 24.762 | >1000[e] | >1000[e] | >1000[e] | >1000[e] |
| Bixafen | 0.850 | 2.625 | 304.650 | >1000[e] | >1000[e] | >1000[e] |
| Boscalid | 1.975 | 226.220 | >1000[e] | >1000[e] | >1000[e] | >1000[e] |
| *Benzimidazoles* | | | | | | |
| Carbendazim | 0.428 | 0.282 | 0.647 | 0.540 | 0.914 | 0.830 |
| Thiophanate | 160.559 | 39.220 | >200[e] | 161.610 | >200[e] | >200[e] |
| Thiabendazole | 1.525 | 1.265 | 2.068 | 1.805 | 2.155 | 2.395 |
| *Multi-site fungicides* | | | | | | |
| Captan[d] | 0.150 | 0.120 | 0.634 | 0.956 | 3.399 | 4.850 |
| Chlorothalonil | 0.128 | 0.095 | 0.835 | 0.610 | 1.475 | 1.210 |
| Mancozeb[d] | 0.695 | 0.850 | 2.915 | 3.136 | 7.150 | 6.450 |
| *Mono-alkyl liopophilic cations (MALCs)* | | | | | | |
| CTAB | 193.620 | 119.275 | 544.300 | 311.650 | 688.550 | 442.250 |
| Dodine | 3.301 | 3.213 | 12.690 | 20.450 | 103.150 | >100[e] |
| C$_{18}$DMS | 319.620 | 115.625 | >1000[e] | 364.215 | >1000[e] | 1000 |
| *"Organic" fungicides* | | | | | | |
| LMW chitosan[d] | 45.650 | 203.060 | 913.850 | 972.750 | >1800[e] | >1800[e] |
| Copper[d] | 110.675 | 66.625 | 202.775 | 186.680 | 264.250 | 299.500 |
| Garlic oil | 35.720 | 50.575 | 107.925 | 166.625 | 163.250 | 243.350 |

All values are given as µg ml$^{-1}$ (ppm)

[a]Graphically determined from non-linear regression curves

[b]EC$_{50}$ and EC$_{90}$ represent concentration at which colony formation is inhibited by 50% and 90%, respectively

[c]MIC (= minimal inhibitory concentration) represents concentration at which no colony formation is detectable

[d]FocTR4 more sensitive than IPO323

[e]No complete inhibition of colony growth detectable

concentrations (Fig 3, e.g. epoxiconazole, pyraclostrobin; see S2 Fig, phase 1 and phase 2). This response indicates a majority of fungicide-sensitive cells and a subpopulation of fungicide-tolerant "persisters". Microscopic investigation revealed that persisters grown on media-containing agar plates represented a mixture of "yeast-like"cells (Fig 2G and 2H, closed arrowheads) as well as thicker, swollen and irregular hyphae (Fig 2G and 2H, open arrowheads). To ensure optimal persister growth conditions, we incubated liquid media cultures of microconidia for 9 days in 3 μg ml$^{-1}$ epoxiconazole, 500 μg ml$^{-1}$ fluxapyroxad or 500 μg ml$^{-1}$ azoxystrobin. Again, FocTR4 survived this treatment (S3A Fig) and formed hyphae (S3B Fig), whereas IPO323 cells died (S3A Fig).

## Azole, SDHI and strobilurin FocTR4 fungicide-tolerance is not based on single-target site mutations

FocTR4 persisters survive high concentration of the global market-leader fungicides, notably azoles, SDHIs and strobilurins [25]. These are single-site fungicides, binding directly to essential pathogen enzymes. Amino acid residue exchanges in the fungicide-interacting region can confer resistance [33,34,35,36]. BLAST searches using published nuclear and mitochondrial FocTR4 genomes [37,38] for putative target genes revealed 3 isoforms of the azole-target lanosterol 14α-demethylase (Foc_Erg11/1, Foc_Erg11/2 and Foc_Erg11/3), homologues of the SDHI-targeted subunits of succinate dehydrogenase (Foc_Sdh2, Foc_Sdh3/1, FocSdh3/2, Foc_Sdh4) and the mitochondrial respiration complex III protein cytochrome b (Foc_Cytb, see S3 Table for UniProt IDs and BLAST error probabilities). We sequenced genomic DNA of epoxiconazole-, fluxapyroxad- or azoxystrobin-treated persisters and searched for mutations in their promoters (1500 bp upstream of translation initiation codon ATG) and in their coding sequences. This revealed no variations between untreated cells and fungicide-grown persisters. However, mutations in unknown transcription factors can confer fungicide resistance (e.g. [24]). We therefore tested if the innate resistance of the newly-generated FocTR4 persister cells is genetically inherited. Persisters were replated onto agar, containing 3 μg ml$^{-1}$ epoxiconazole, 500 μg ml$^{-1}$ fluxapyroxad or 500 μg ml$^{-1}$ azoxystrobin and their growth was compared with fungicide naive cells. We found that fungicide sensitivity of persister cells was the same as untreated wild-type cells (Fig 2I) and conclude that the ability of FocTR4 to cope with high fungicide loads of azoles, SDHIs and strobilurins is not genetically inherited.

## Tolerance of persisters against azoles and SDHIs correlates with high target genes expression

Fungicide resistance can be achieved by elevated target gene expression [39]. This can be pronounced if the target gene carries mutations that affect fungicide binding [40]. We asked if any identified target enzyme-encoding genes carry critical mutations that could prevent epoxiconazole, fluxapyroxad or azoxystrobin from target enzyme inhibition. Residues that impair fungicide binding are known in *C. albicans* lanosterol 14α-demethylase Ca_Erg11 [35,41], *Z. tritici* succinate dehydrogenase subunits [36], and the quinol oxidation site of cytochrome b [33,34]. We compared their predicted protein sequences with FocTR4 homologues and searched for mutations in residues that are (i) implied in fungicide-binding (ii) absent from fungicide-sensitive *Z. tritici* target proteins. This revealed residue substitutions in the fungicide-binding region of all FocTR4 target genes (Fig 4A–4C). The impact of these alterations was assessed using the SIFT web server (see S1 Table for URL and reference), which predicts the consequence of point mutations for protein function. This analysis classified all amino acid exchanges in Foc_Cytb, Foc_Sdh2 and Foc_Sdh4 as "tolerated", suggesting that they do not significantly alter fungicide-binding to these target proteins (Fig 4B and 4C). However,

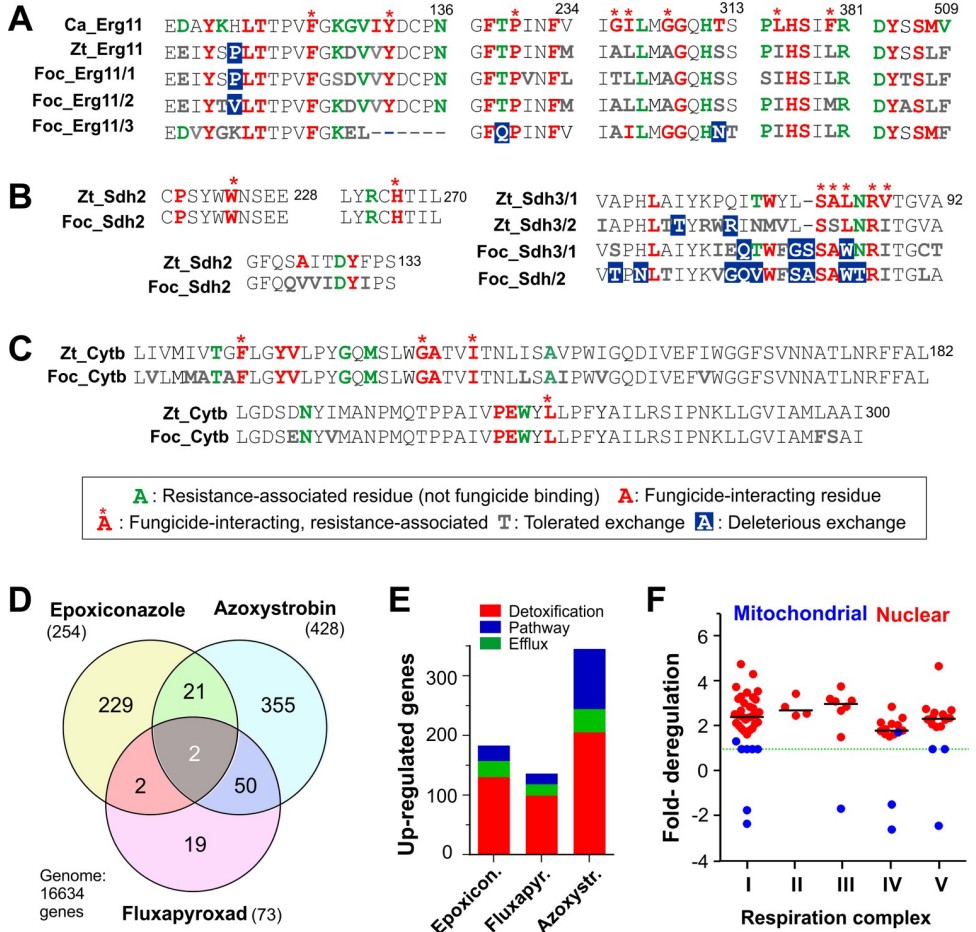

**Fig 4. Fungicide binding sites in target enzymes and fungicide-induced transcriptional changes in FocTR4 persisters. A** Azole binding site in *Candida albicans* Erg11 (Ca_ERG11, UniProt ID: P10613), *Z. tritici* Erg11 (Zt_Erg11, UniProt ID: F9XG32), FocTR4 Erg11_1 (Foc_Erg11_1; UniProt ID: X0JWG9), FocTR4 Erg11_2 (Foc_Erg11_2, UniProt ID: X0LR56) and FocTR4 Erg11_3 (Foc_Erg11_3, UniProt ID: X0JM02). Residues known in *C. albicans* Cyb51, reported to bind the azoles posaconazole and oteseconazole [41], are shown in red-bold; asterisk indicate potential involvement in resistance [35]; the impact of all residue substitutions in the *Z. tritici* and FocTR4 homologues of Ca_ERG11 was estimated using the SIFT server (https://sift.bii.a-star.edu.sg/); sequence alignment and determination of conserved amino acid exchanges used Clustal Omega (https://www.ebi.ac.uk/Tools/msa/clustalo/). **B** SDHI-binding site in succinate dehydrogenase subunits of *Z. tritici* (ZtSdh2, UniProt ID: O42772; ZtSdh4, UniProt ID: F9X9V6; ZtSdh3_1, UniProt ID: F9XH52; ZtSdh3_2, UniProt ID: F9XLX9 (partial); complete ZtSdh3_2 sequence at FungiDB, https://fungidb.org/fungidb/app/record/gene/ZTRI_10.476;) and FocTR4 (FocSdh2, UniProt ID: X0JUP8; FocSdh4, UniProt ID: X0JC07; FocSdh3_1, UniProt ID: X0JTM7; FocSdh3_2, UniProt ID: X0K132). Residues known in *Z. tritici* and other fungi to bind SDHI molecules [36] are shown in red-bold; asterisks indicate potential involvement in resistance [36]; the impact of all residue substitutions in FocTR4 homologues was estimated using the SIFT server (https://sift.bii.a-star.edu.sg/); sequence alignment used Clustal Omega (https://www.ebi.ac.uk/Tools/msa/clustalo/). **C** Comparison of the amino acid sequence of quinol oxidation (Qo) site of mitochondrial cytochrome b in *Z. tritici* (ZtCytb; UniProt ID: Q6X9S4) and FocTR4 (FocCytb, UniProt ID: A0A1A7TD23). Residues known to bind strobilurin molecules are shown in red-bold; asterisks indicate potential involvement in resistance; the impact of all residue substitutions in FocTR4 was estimated using the SIFT server (https://sift.bii.a-star.edu.sg/); sequence alignment used Clustal Omega (https://www.ebi.ac.uk/Tools/msa/clustalo/). Note that all substitutions in FocCytb are predicted to be of low impact ("tolerated"). Information on strobilurin molecule-interacting (Cpd-interacting) and resistance-conferring residues from [33,34]. **D** Venn diagram showing numbers of FocTR4 genes, up-regulated significantly (>5-fold) in the presence of fungicides. Note that a transcription factor (UniProt ID: X0JJ38) and an ABC transporter (UniProt ID: X0JJA9) are up-regulated in all conditions. **E** Number of up-regulated genes in FocTR4 persister cells, putatively involved in providing fungicide tolerance. Putative detoxifying enzymes (red), efflux transporter and pumps (green) and proteins involved in the targeted cellular pathway (blue) that are significantly upregulated (P<0.05) in epoxiconazole, fluxapyroxad and azoxystrobin-persisters are included. **F** Regulation of FocTR4 mitochondrial respiration complex subunits in the presence of 500 μg ml⁻¹ azoxystrobin. Dotted green line indicates non-significant difference to methanol-treated control cells. Median of expression of nucleus-encoded subunits (red) is indicated by

black lines. Data shown in (**D**) were derived from RNA-preparations of cells, grown for 2 days in medium, supplemented with 3 μg ml$^{-1}$ epoxiconazole, 500 μg ml$^{-1}$ azoxystrobin or 500 μg ml$^{-1}$ fluxapyroxad.

Foc_Erg11/3 showed 2 unique residue substitutions, classified as "deleterious" (Fig 4A) in positions associated with azole-tolerance in *C. albicans* [41] (T208Q, SIFT score 0.02), or predicted to affect azole molecule binding (T289N, SIFT score 0.02). Moreover, the SDHI-binding region of subunits Foc_Sdh3/1 and particularly Foc_Sdh3/2 were significantly altered. Both carried a residue 82 insertion and a point mutation in a SDHI-interacting residue (L84W, Fig 4B), with Foc_Sdh3/2 showing 2 "deleterious" substitutions in positions associated with SDHI-tolerance (T77V, N85T, SIFT score 0.02 for both). These results, collectively, suggest that increased expression of Foc_Erg11/3 and Foc_Sdh3/2 could participate in azole and SDHI resistance, respectively.

We tested this hypothesis by comparing the transcriptome of epoxiconazole and fluxapyroxad persister hyphae with that of untreated control hyphae, both grown from microspores, over 2 days. We found that the gene encoding Foc_Sdh3/2 was strongly up-regulated in fluxapyroxad persister cells (~88-fold, Tables 2 and S4). A comparable up-regulation of the homologous subunit Zt_Sdh3/2 was found in fluxapyroxad-treated IPO323 cells (S4 Fig, S5 and S6 Tables). However, Zt_Sdh3/2 carried only few "deleterious" exchanges in non-critical residues of the SDHI-binding region (Fig 4B). Consequently, this subunit should still bind SDHI molecules; this may explain the lethal effect of fluxapyroxad on *Z. tritici* cells. Thus, it is likely that increased expression of the SDHI-insensitive succinate dehydrogenase subunit Foc_Sdh3/2 contributes to innate SDHI tolerance in persister cells.

We investigated expression of genes encoding the 3 isoforms of lanosterol 14α-demethylase (Foc_Erg11/1-3). Whilst *Foc_erg11/3* was not up-regulated in epoxiconazole persister cells, the azole target genes *Foc_erg11/1* and *Foc_erg11/2* were strongly induced (~661-fold and ~24-fold, respectively, Tables 2 and S6). Up-regulation of fungal lanosterol 14α-demethylase-encoding genes is a common response to azole treatment. However, the fold-change ranges from 2.5- to maximal 58-fold [42,43,44]. To test if the unusually high increase of expression is a FocTR4-specific adaptation, or due to technical variations, we determined the transcriptome of fungicide-sensitive *Z. tritici* cells treated with 3 μg ml$^{-1}$ epoxiconazole, 500 μg ml$^{-1}$ fluxapyroxad, or 500 μg ml$^{-1}$ azoxystrobin, harvested after 2 d, when 60–80% of all cells were still alive (S3A Fig). We found up-regulation of an Erg11 homologue (Zt_Erg11, UniProt: F9XG32), yet at much lower fold-change (~6.6-fold, S5 and S6 Tables). This suggests that strong expression of Erg11 genes is a FocTR4-specific adaptation that could participate in azole tolerance.

## Non-target site mechanisms may participate in FocTR4 fungicide tolerance to azoles, SDHIs and strobilurins

Fungi can acquire fungicide resistance by increased synthesis of resistance-associated proteins [22]. These non-target site mechanisms include efflux transporters, detoxifying enzymes, such as cytochrome P450 monooxygenases, or components of the targeted biosynthetic pathway [45,46,39,47,48]. We investigated if such genes are up-regulated in fungicide-tolerant persister cells and focussed on genes >5-fold induced and that encode putative (i) xenobiotic detoxifying enzymes (including, amongst others, cytochrome P450 and other monooxygenases, dehydrogenases, oxidases, esterases, reductases, glucuronosyltransferases, sulfotransferases, glutathione S-transferases [49,50]), (ii) fungicide efflux pumps and transporters (iii) enzymes participating in the targeted pathway. This analysis identified 2 genes strongly-expressed in the presence of epoxiconazole, fluxapyroxad and azoxystrobin (Fig 4D, grey sector; S4 Table).

**Table 2. Strongly induced genes, putatively involved in fungicide tolerance in FocTR4.**

| Enzyme[a] | Induction[b] | Identifier[c] | Homologue[d] | BLAST[e] |
|---|---|---|---|---|
| **Azole** | | | | |
| | | | *Enzymatic detoxification* | |
| Glutathione S-transf. | 20.79 | X0LFB8 | Glutathione S-transferase Gtt1p, *S. cerevisiae* | 2e-10 |
| Monooxygenase | 20.03 | X0IQF2 | Hypothetical protein, *N. crassa* | 3e-55 |
| Monooxygenase | 15.53 | X0JX05 | Alkanesufonate monooxygenase., *N. crassa* | 0.0 |
| Monooxygenase | 12.21 | X0K5V1 | Flavin-binding monooxygenase, *N. crassa* | 3e-122 |
| Cytochrome P450 | 102.32 | X0ITB1 | Oxysterol 7α-hydroxylase, human | 1e-09 |
| Cytochrome P450 | 203.01 | X0J7G5 | 24-hydroxycholesterol 7-α-hydroxylase, human | 7e-07 |
| | | | *Transport over membranes* | |
| ABC transporter | 232.46 | X0JJA9 | Multidrug transporter MDR3, *T. rubrum*[g] | 0.0[e] |
| MFS transporter | 153.61 | X0IWA4 | MCH4 transporter, *S. cerevisiae* | 6e-41[e] |
| | | | *Target gene up-regulation* | |
| Erg11_1 | 661.01 | X0JWG9 | Lanosterol demethylase Erg11p, *S. cerevisiae* | 1e-162 |
| Erg11_2 | 23.74 | X0LR56 | Lanosterol demethylase Erg11p, *S. cerevisiae* | 1e-169 |
| | | | *Pathway alteration* | |
| Erg6_1 | 462.52 | X0JEW4 | Sterol 24-C-methyltransferase, *S. cerevisiae* | 2e-127 |
| Sim1 | 131.24 | X0JVY8 | 3β-hydroxysteroid-Δ(8)Δ(7)isomerase, human[h] | 5e-47 |
| Erg13 | 12.74 | X0K3N0 | hydroxymethylglutaryl-CoA synth., *S. cerevisiae* | 0.0 |
| Erg25_1 | 13.73 | X0J3L6 | C-4 methylsterol oxidase, *S. cerevisiae* | 2e-126 |
| **SDHI** | | | | |
| | | | *Enzymatic detoxification* | |
| Glutathione S-transf. | 13.97 | X0JN47 | No hit in *S. cerevisiae*, human or *N. crassa* | - |
| Dehydrogenase | 57.71 | X0JTQ9 | Dehydrogenase./reductase DHRS7B, human | 1e-25 |
| | | | *Transport over membranes* | |
| ABC transporter | 14.10 | X0JJA9 | Multidrug transporter MDR3, *T. rubrum*[g] | 0.0[e] |
| MFS transporter | 13.19 | X0J859 | EfpA transporter, *M. tuberculosis*[i] | 4e-21[e] |
| | | | *Target gene up-regulation* | |
| Sdh3_2 | 88.31 | X0K132 | Succinate dehydrogenase sub. Shh3p, *S. cerevisiae* | 1e-09 |
| **Strobilurin** | | | | |
| | | | *Enzymatic detoxification* | |
| Acetyltransferase | 19.44 | X0LJ02 | Hypothetical protein, *N. crassa* | 4e-06 |
| Dehydrogenase | 21.24 | X0JM78 | Tetrahydroxynaphthalene reduct., *N. crassa* | 1e-31 |
| Dehydrogenase | 21.07 | X0JRJ4 | Alcohol dehydrogenase Adh6p, *S. cerevisiae* | 1e-102 |
| Dehydrogenase | 19.56 | X0J169 | Retinol dehydrogenase 12, human | 5e-45 |
| Dehydrogenase | 16.04 | X0JDC0 | Aldehyde dehydrogenase Ald5p, *S. cerevisiae* | 4e-82 |
| Dehydrogenase | 12.11 | X0KKG1 | No hit in *S. cerevisiae*, human or *N. crassa* | - |
| Dehydrogenase | 11.82 | X0JR95 | Reticulon-4-interacting protein, human | 3e-17 |
| Dehydrogenase | 10.68 | X0JW86 | Sort chain oxidoreductase, *N. crassa* | 3e-13 |
| | | | *Transport over membranes* | |
| MFS transporter | 60.36 | X0J7L3 | MCH4 transporter, *S. cerevisiae* | 3–54[e] |
| MFS transporter | 42.49 | X0J859 | EfpA transporter, *M. tuberculosis*[i] | 4e-21[e] |
| MFS transporter | 23.19 | X0J5S1 | MCH2 transporter, *S. cerevisiae* | 3e-75[e] |
| MFS transporter | 12.77 | X0J3S0 | MCH4 transporter, *S. cerevisiae* | 8–24[e] |
| | | | *Oxidative stress proteins* | |
| Cytochro. c peroxidase | 18.77 | X0KC05 | Cytochrome c peroxidase Ccp1p, *S. cerevisiae* | 4e-06 |

(*Continued*)

**Table 2.** (Continued)

| Enzyme[a] | Induction[b] | Identifier[c] | Homologue[d] | BLAST[e] |
|---|---|---|---|---|
| Catalase | 17.70 | X0K1K8 | Catalase Cta1p, *S. cerevisiae* | 7e-106 |

All entries represent genes that are upregulated >10-fold in the presence of 3 μg ml$^{-1}$ epoxiconazole, 500 μg ml$^{-1}$ fluxapyroxad or 500 μg ml$^{-1}$ azoxystrobin

[a]Predicted enzyme category

[b]Average fold-upregulation over respective methanol control; [c]UniProt gene identifier at https://www.uniprot.org/

[d]human homologue shown when no hit in *S. cerevisiae*; *N. crassa* homologue shown when no hit in human and yeast

[e]P-values for BLAST search done at NCBI (https://blast.ncbi.nlm.nih.gov/ Blast.cgi?PAGE = Proteins)

[f]BLAST search done at "transporter Classification Database" (http://www.tcdb.org)

[g]Also homologous to multidrug resistance protein CDR1/2 in *Candida albicans* (BLAST P = 0.0) and AtrB in *Aspergillus nidulans* (BLAST P = 0.0)

[h]Involved in cholesterol biosynthesis [54]

[i]Putative multi-drug resistance pump in *Mycobacterium tuberculosis* [60]; for more information see S4, S5 and S6 Tables

Firstly, we found an ABC transporter (UniProt: X0JJA9), highly homologous to multi-drug efflux pump Mdr3 in *Trichophyton rubrum* [51] (BLAST search at http://www.tcdb.org, P = 0.0), and secondly, a putative transcription factor (UniProt ID: X0JJ38), with weak similarity to an uncharacterised transcription factor in *Botrytis cinerea* (Cys6; BLAST P = 7e-14, UniProt ID: M7TPG4).

Most other transcriptional responses were fungicide-specific. Epoxiconazole induced ~130 candidate enzymes, implied in detoxification (Fig 4E; S4 Table), with 6 genes being expressed >12 -fold (Table 2). Amongst these are 2 cytochrome P450 enzymes with no homologues in *S. cerevisiae*, but significant similarity to human oxysterol 7α-hydroxylase (CYP7B1, BLAST P = 1e-09) and 24-hydroxycholesterol 7α-hydroxylase (CYP39A1, BLAST P = 4e-07), both involved in cholesterol metabolism [52]. In addition, FocTR4 strongly-induced an ABC transporter (~230-fold, Table 2), an MFS transporter (150-fold), and, to a lesser extent, several RTA1-like lipid-translocating exporters, thought to remove aberrant sterol intermediates from the cell [53] (Table 2, S4 Table). This suggests that several detoxification mechanisms contribute to azole tolerance in FocTR4 persisters. We also found alterations in expression of ergosterol biosynthesis enzymes. Of the 37 *erg* genes, putatively involved in the ergosterol biosynthesis, 23 were significantly up-regulated, including all genes encoding enzymes in the late ergosterol biosynthesis pathway (S5 Fig; S5 and S6 Tables). The strongest increase in gene expression was found for Foc_Erg6/1 (~460-fold, Tables 2 and S6) and a putative 3-β-hydroxysteroid-Δ(8),Δ(7)-isomerase (~130-fold, named Sim1, Table 2). Sim1 lacks a homologue in *S. cerevisiae* but shows significant similarity to a human enzyme involved in cholesterol biosynthesis [54] (BLAST P = 5e-47). Homologues of *erg6* and *sim1* in *Z. tritici* were also induced, but to a much lesser extent (~7.1-fold and ~11.2-fold, S5 and S6 Tables). The precise role of Foc_Erg6 or Foc_Sim1 in persister cells is unknown, but their high expression maybe indicative of alterations in the ergosterol biosynthesis pathway. Collectively, the combined action of detoxifying enzymes, efflux pumps, ergosterol-biosynthesis enzymes further increase the ability of persister cells to cope with azole fungicides.

We investigated transcriptional responses of FocTR4 persisters and IPO323 to SDHIs. Fluxapyroxad-tolerant FocTR4 persisters showed moderate induction of putative efflux pumps (one ABC transporter at ~14-fold, one MFS transporter at ~13-fold) and only one strongly induced putative dehydrogenase (~58-fold, Tables 2 and S6). In contrast, IPO323 cells, which

largely perished in fluxapyroxad (S3A Fig), showed strong up-regulation of efflux pumps (*e.g.* an ABC transporter, ~70-fold, 4 MFS transporters ~140 to 390-fold) and putative detoxification cytochrome P450 enzyme-encoding genes (the two most strongly induced genes reach ~310- and ~1180-fold up-regulation, S5 and S6 Tables). Thus, up-regulation of the aberrant Foc_Sdh3/2 subunit may underpin SDHI-tolerance in FocTR4, but this merits further investigation.

Finally, we interrogated the nuclear and mitochondrial transcriptomes of azoxystrobin-tolerant FocTR4 cells, asking if *cytb*, encoding the mitochondrial target protein cytochrome b, is induced in persisters. This revealed that the strobilurin-target gene *Foc_Cytb* was down-regulated (~1.7-fold, S7 Table), which suggests that strobilurin-tolerance in persisters is not due to elevated target enzyme production. Moreover, while all nuclear-encoded components of the respiration complexes I-IV were upregulated, almost all core mitochondrial respiratory chain proteins were either unaltered in expression or down-regulated in azoxystrobin (Fig 4F, S7 Table). As functional respiration complexes require all core genes, it is unlikely that upregulation of the respiration chain underpins FocTR4 azoxystrobin tolerance.

We asked if there are indications for non-target site mechanisms in azoxystrobin-treated FocTR4 cells and found upregulation of 211 genes encoding putative detoxifying enzymes, with 8 genes induced ~10-21-fold (Table 2). In addition, 4 putative MFS transporters were upregulated (~13-60-fold, Table 2). Moreover, expression of potential mitochondrial antioxidants was induced in azoxystrobin-treated FocTR4, but not in IPO323 cells (*e.g.* cytochrome c peroxidase, ~19-fold; catalase, ~18-fold; Tables 2 and S6). In contrast, an alternative oxidase (UniProt ID: X0KNM7; similar to *N. crassa* AOD1, BLAST P = 6e-169), known in other fungi to protect against oxidative stress induced by strobilurins [55], was surprisingly down-regulated (~5.7-fold, S6 Table). Most of the induced antioxidant proteins are predicted to locate in mitochondria (TargetP-2.0 prediction, see S6 Table) and/or have homologues in *S. cerevisiae*, where their mitochondrial localisation was demonstrated ([56,57], S6 Table). This suggests that FocTR4 persisters detoxify surplus electrons within mitochondria.

Strobilurins bind to Cytb, resulting in the blockage of electron transfer at complex III. This may lead to electron leakage and production of toxic reactive oxygen species (ROS) that trigger apoptotic fungal cell death (Fig 5A, [58]). We tested if (i) azoxystrobin induces ROS in strobilurin-sensitive IPO323 cells (ii) FocTR4 shows reduced ROS development, as assayed with the superoxide-sensitive dye dihydrorhodamine 123 (DHR123, [59]). This revealed that azoxystrobin increased ROS levels in IPO323, but not in FocTR4 persisters (Fig 5B and 5C). We tested if high levels of ROS induce apoptotic fungal cell death in IPO323, by identifying early apoptotic cells using the green-fluorescent reporter FITC-VAD(OMe)-FMK [58]. Consistent with increased ROS, IPO323 cells underwent apoptosis, whereas FocTR4 showed no significant increase in apoptotic cell numbers (Fig 5D and 5E). These results support the notion that azoxystrobin persisters effectively detoxify mitochondrial ROS, thereby preventing oxidative damage and programmed cell death.

In mitochondria, electron transport through the respiration chain polarises the inner membrane, which is required to synthesise ATP (Fig 5A). We speculated that azoxystrobin-induced blockage of electron transport at complex III inhibits this process. Quantitative analysis of tetramethylrhodamine methyl ester (TMRM) fluorescent signal intensities revealed that azoxystrobin treatment significantly reduced mitochondrial potential in both fungi (Fig 5F). However, even in the presence of 500 µg ml$^{-1}$ azoxystrobin, FocTR4 persister cells maintained 66.1 ± 4.0% of their mitochondrial polarization, which is significantly more than in IPO323 cells at 35.5 ± 3.1% (Fig 5F). Thus, FocTR4 persisters maintain >50% of their proton motive force, suggesting continuous ATP production. Our transcriptome data did not provide any clear indication of how this is achieved. However, it is worth noting that azoxystrobin-

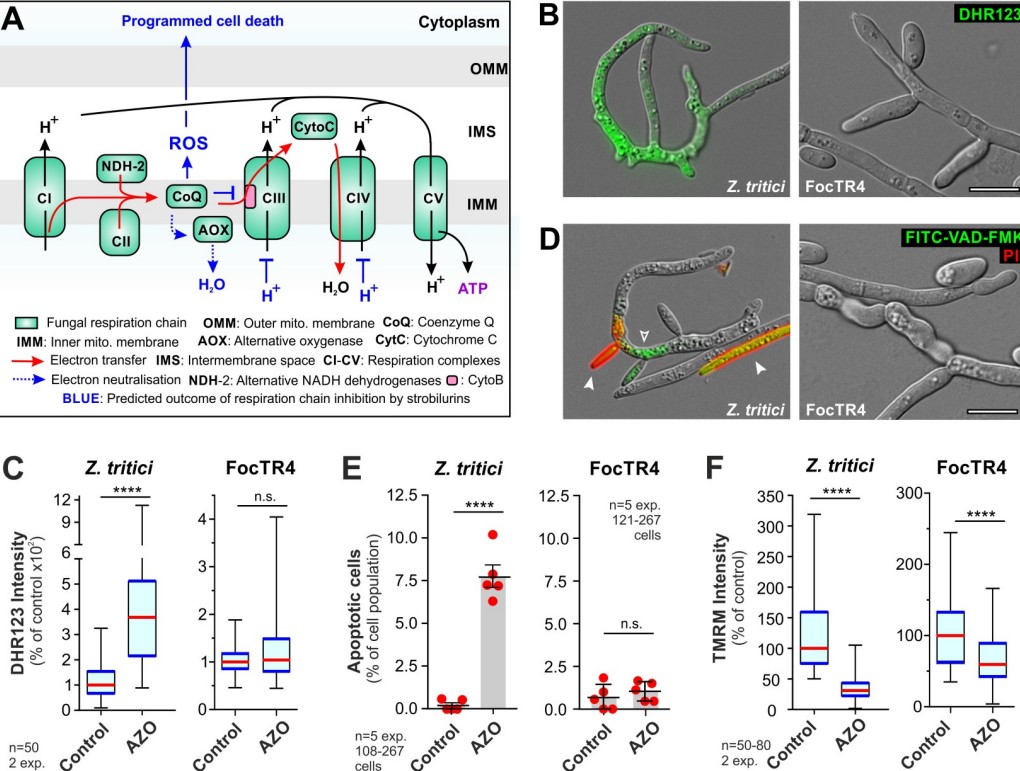

**Fig 5. The effect of azoxystrobin on *Z. tritici* IPO323 and FocTR4. A** Predicted effect of strobilurin on fungal respiration. Electrons are generated at complexes I and II, and by alternative NADH dehydrogenases. Coenzyme Q delivers them cytochrome b (Cytb) at complex III, from where they pass to cytochrome C (CytC) and arrive at complex IV (CIV) for neutralisation. This generates a proton gradient for ATP synthesis. Strobilurins block electron transfer Cytb (blue blunt arrow), which could form reactive oxygen species (ROS), known to trigger fungal apoptotic cell death [58]. Strobilurin-induced ROS was reported to be neutralised by fungal alternative oxidase (AOX, [55]), yet this enzyme is down-regulated in strobilurin persister cells. **B** Reactive oxygen species in *Z. tritici* IPO323 and FocTR4 cells, visualised with DHR123 (green). Both cell types were treated with 500 μg ml⁻¹ azoxystrobin for 24 h. Scale bar = 10 μm. **C** ROS levels in azoxystrobin-treated *Z. tritici* and FocTR4 cells, visualised by DHR123 staining. **D** Apoptotic cells in *Z. tritici* IPO323, visualised with FITC-VAD(OMe)-FMK (green, open arrowhead). No apoptotic cells were found in FocTR4. Both cell types were treated with 500 μg ml⁻¹ azoxystrobin for 24 h. Dead cells appear yellow due to co-staining with red-fluorescent propidium iodide (closed arrowhead). Note that these cells were not included in (**E**). Scale bar = 10 μm. **E** Apoptotic cells after treatment with azoxystrobin in *Z. tritici* and FocTR4, visualised by FITC-VAD(OMe)-FMK staining. **F** Mitochondrial membrane potential in azoxystrobin-treated *Z. tritici* and *FocTR4* cells, treated with 500 μg ml⁻¹ azoxystrobin and visualised by staining with the voltage-sensitive dye TMRM.FocTR4 persister cells maintained 66.1 ±4.0% (n = 80) of their mitochondrial polarization. IPO323 cells 35.5±3.1% (n = 50 cells). In (**C—F**) 500 μg ml⁻¹ azoxystrobin was applied for 24 h. Control indicates the use of an equivalent volume of the solvent methanol. Non-normally distributed data in (**C, F**; Shapiro-Wilk test, P<0.05) shown as Whiskers' plots with 25/75 percentiles (blue lines) and median (red lines); bars in (**E**) represent mean ± SEM, red dots represent independent experiments; sample sizes indicated; statistical analysis used non-parametric Mann-Whitney testing (**C, F**), or Student's t-testing with Welch correction (**E**); n.s. = non-significant difference to control at two-tailed P = 0.2144 (**C**) and P = 0.4264 (**E**); **** = significant difference to control at two-tailed P<0.0001 (**C, E, F**).

treatment strongly-induces 38 proteins of unknown function (~10 to-165-fold), with 14 of these carrying a putative secretion signal (TargetP-2.0 prediction, S8 Table). Future studies are needed to elucidate if these proteins participate in strobilurin tolerance in FocTR4.

## All three FocTR4 spore forms survive treatment with most fungicides

So far, our fungicide assays were performed in nutrient-rich medium, where persister hyphae survived azole, SDHI and strobilurin treatment. To better reflect the nutrient-poor soil

environment, fungicide toxicity on chlamydopores, microspores and macrospores was assessed after 10 days in sterile distilled water (SDW), using quantitative LIVE/DEAD staining (Fig 6A shows examples). We also attempted to investigate hyphae in these assays. However, even in the absence of fungicide hyphae perished within 3–10 days in water (~60% dead after 10 days, not shown). Consequently, we only incubated all 3 spore forms with the highly effective fungicides (MIC<10 μg ml$^{-1}$) at ~5-times of their MIC (carbendazim, 4.5 μg ml$^{-1}$; thiabendazole, 10 μg ml$^{-1}$; chlorothalonil, 5 μg ml$^{-1}$; captan, 10 μg ml$^{-1}$; mancozeb, 35 μg ml$^{-1}$) and all less effective chemistries (MIC>10 μg ml$^{-1}$) at 100 μg ml$^{-1}$. All single-site fungicides, including azoles, strobilurins, SDHIs and benzimidazoles were largely ineffective in killing all 3 FocTR4 spore types; only tebuconazole and pyraclostrobin caused modest mortality in macro- and microconidia, but not in chlamydospores (Fig 6B–6D).

Interestingly, spores suspended in SDW culture were not killed by the single-site fungicides carbendazim and thiabendazole, or by the multi-site compound chlorothalonil (Fig 6B–6D). As these fungicides effectively inhibit growth on agar plates, we conclude that these compounds are fungistatic and not fungitoxic. Only the multi-site fungicides mancozeb, copper and three MALCs effectively killed all micro-, macroconidia and chlamydospores (Fig 6B–6D). This included CTAB and C$_{18}$DMS, which showed lower effectiveness in agar plates, but greater efficacy in liquid culture. We also noted that several multi-site fungicides were less effective against chlamydospores (Fig 6B–6D, e.g. captan and LMW chitosan). As these durable soil-surviving spores initiate infection [3], a more detailed analysis was performed. This revealed that chlamydospores are enveloped by a fibrillar outer layer and an inner multi-layered wall (S6A Fig). This laminated wall is significantly thicker than in other morphotypes (S6B Fig) and could reduce fungicide efficacy. Finally, we tested if azole-, SDHI- and strobilurin-persisting spores undergo a morphological transition and found that surviving spores after 10 days incubation with 100 μg ml$^{-1}$ still display their characteristic morphologies (Fig 6E), suggesting that these spores are innately resistant to these fungicides. In summary, our data show that all 12 tested single-site and 1 multi-site fungicide show no, or low, fungitoxicity in spores. This includes the major fungicide classes azoles, SDHIs and strobilurins.

## MALCs protect banana plants against Panama disease

Our experiments demonstrating that the multi-site fungicides mancozeb, copper and 3 MALCs (CTAB, dodine and C$_{18}$DMS) were fungitoxic to all 3 spore types in SDW culture, propelled us to evaluate these fungicides in banana virulence assays. In addition, we included captan and LMW chitosan, which effectively killed micro- and macroconidia in liquid culture, were potent at inhibiting growth on agar plates, or were reported to protect against Panama disease [20]. To enhance effectiveness of all fungicides in soil, the lethal concentrations, determined in liquid culture fungicide mortality assays, were doubled (see Figure legends and Material and Methods) and supplemented the fungicide solutions with the wetting agent Silwet L-77.

We first tested if fungicides are phytotoxic, by applying droplets (10 μl) to attached banana leaves, followed by monitoring tissue damage after 24 h at 27˚C. Droplets of 50 mg ml$^{-1}$ benzalkonium chloride (BAC), used as positive control, caused leaf necrosis (S7A Fig; arrowheads indicate area covered by droplet), whereas none of the fungicide solutions caused leaf damage (S7A Fig). Next, we tested if root-application of the fungicides affected plant health and thus watered 8 week-old plants, at 27˚C, with the 7 fungicide solutions twice (as described in Materials and Methods), at the same concentrations, and used 50 mg ml$^{-1}$ BAC as a positive control. Again, BAC treatment caused plant damage after 21 d, whereas fungicide-treated plants showed no obvious health differences from plants treated with water (Control (-) = water, Control (+) = BAC; S7B Fig).

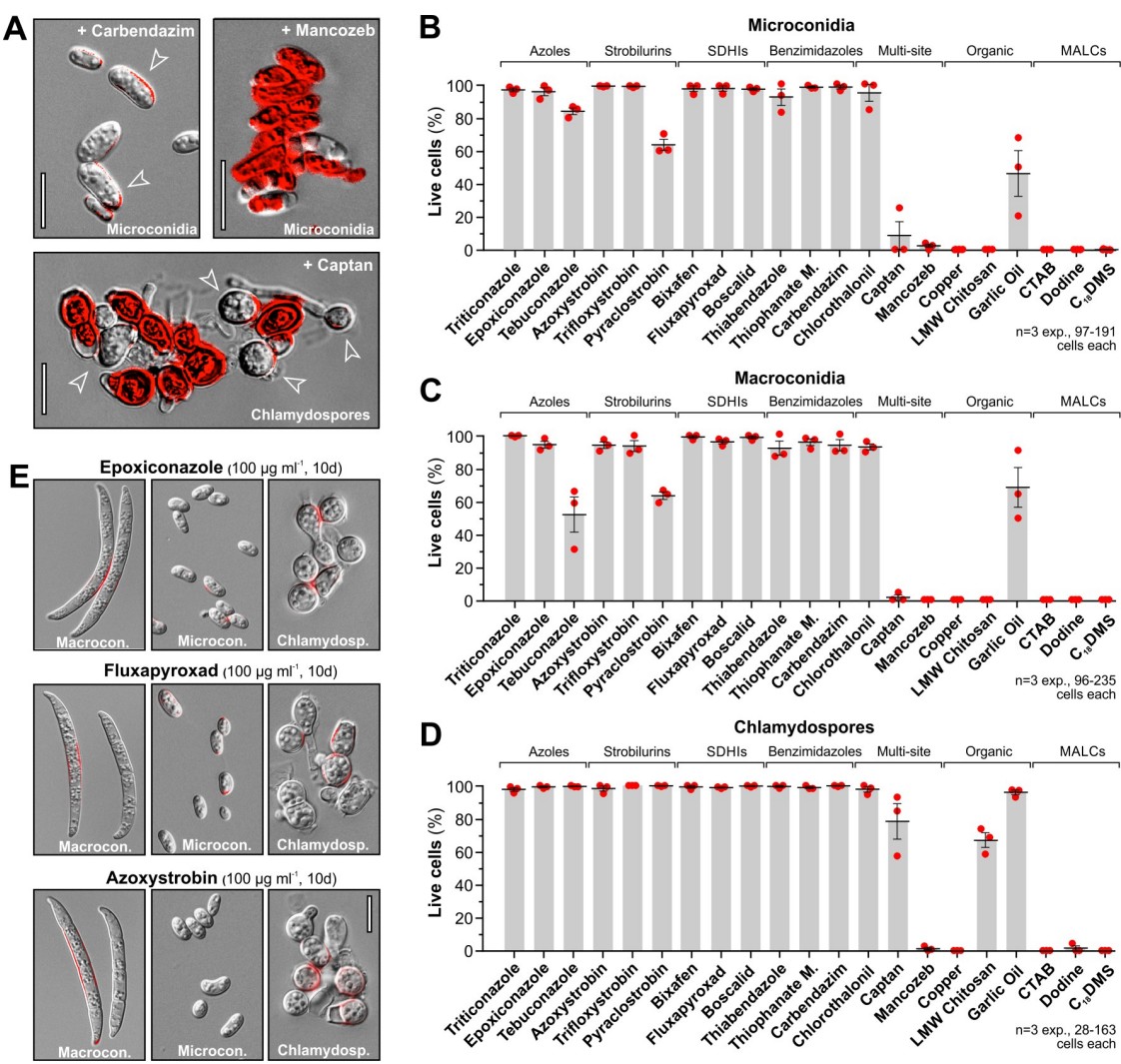

**Fig 6. Fungicide sensitivity of FocTR4 spores in liquid culture. A** Examples of LIVE/DEAD dye-stained microconidia and chlamydospores after 10 days in ddH$_2$O, supplemented with carbendazim, mancozeb or captan. Living cells exclude the dye and often accumulate it in the cell wall (open arrowheads); dead cells fluoresce red. Scale bars = 10 μm. **B** Relative number of living microconidia (LIVE/DEAD staining negative) after 10 d-treatment with fungicides; fungicide classes are indicated above parenthesis. **C** Relative number of living macroconidia (LIVE/DEAD staining negative) after 10 d-treatment with fungicides; fungicide classes are indicated above parenthesis. **D** Relative number of living chlamydospores (LIVE/DEAD staining negative) after 10 d-treatment with fungicides; fungicide classes are indicated above parenthesis. **E** Morphology and LIVE/DEAD staining of microconidia, macroconidia and chlamydospores after 10 day-incubation in SDW, supplemented with epoxiconazole, fluxapyroxad and azoxystrobin (all 100 μg ml$^{-1}$). Note that all spore forms can persist at high concentration of fungicides. Scale bar = 10 μm. Bars shown in (**B—D**) represent the mean proportion (±SEM) of cells that did not take up LIVE/DEAD staining, thus were considered alive; red dots represent averages of 3 independent experiments. All spore types in (**B—D**) were incubated for 10 days in SDW at 25°C and under rotation; concentrations used were in category I (= inhibition on agar plates at <10 μg ml$^{-1}$): thiabendazole, 10 μg ml$^{-1}$; carbendazim, 4.5 μg ml$^{-1}$; chlorothalonil, 5 μg ml$^{-1}$; captan, 10 μg ml$^{-1}$; mancozeb, 35 μg ml$^{-1}$; in category II (= inhibition on agar plates at >10 μg ml$^{-1}$): copper, 100 μg ml$^{-1}$ (applied as Copper(II) sulfate pentahydrate); LMW chitosan, 100 μg ml$^{-1}$ (applied as a lactate salt); garlic oil, 100 μg ml$^{-1}$; CTAB (Cetrimonium bromide), 100 μg ml$^{-1}$; dodine, 100 μg ml$^{-1}$; C$_{18}$DMS; 100 μg ml$^{-1}$; triticonazole, 100 μg ml$^{-1}$; epoxiconazole, 100 μg ml$^{-1}$: tebuconazole, 100 μg ml$^{-1}$; azoxystrobin, 100 μg ml$^{-1}$; trifloxystrobin, 100 μg ml$^{-1}$; pyraclostrobin, 100 μg ml$^{-1}$; bixafen, 100 μg ml$^{-1}$; fluxapyroxad, 100 μg ml$^{-1}$; boscalid, 100 μg ml$^{-1}$; thiophanate methyl, 100 μg ml$^{-1}$; concentrations in (**e**) are 100 μg ml$^{-1}$.

Next, we tested all 7 fungicides for their ability to protect plants from disease. Chlamydospores are most field-relevant [3] and thus were used in these experiments. At 1 h post-inoculation of 8–12 week old banana plants, fungicides were applied (day 0) and again applied 7

days later. Disease symptoms were assessed at 56 dpi (see Materials and Methods for details). Chlamydospores alone cause wilting and necrosis of aerial tissues (Fig 7A, Control (+)) and induced necrosis of the corm (Fig 7B, Control (+)). Treatment with the various fungicides resulted in banana plants with graduated disease symptoms (Fig 7A–7C). Mancozeb and

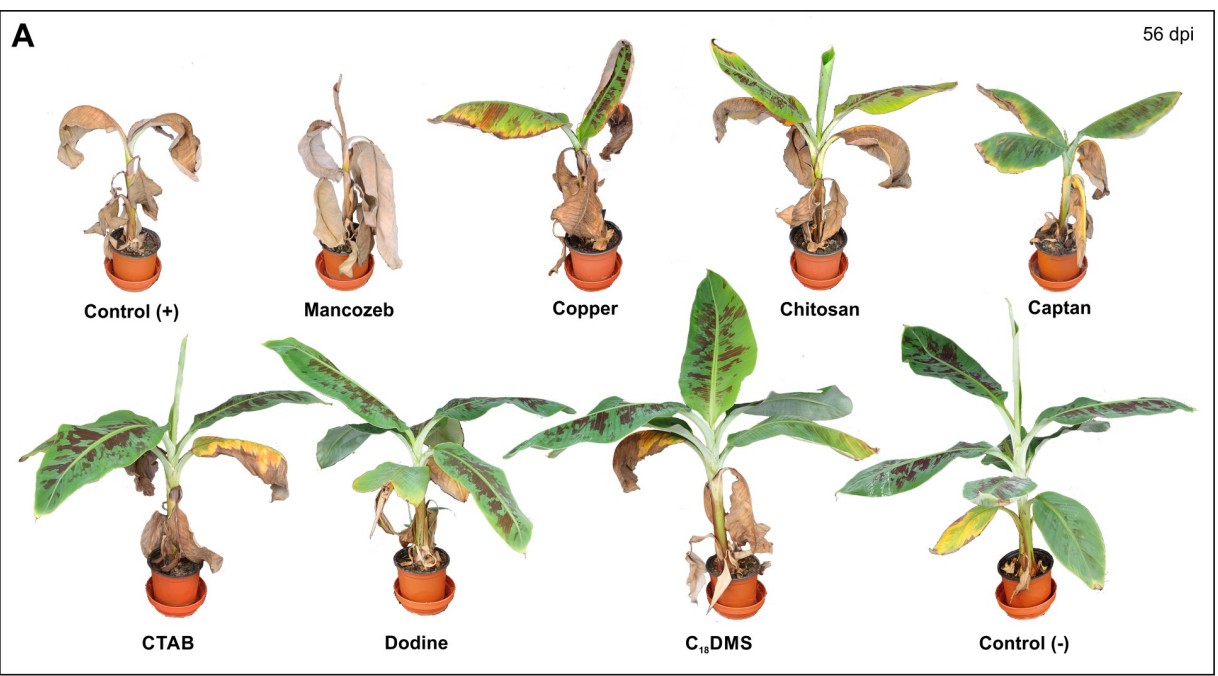

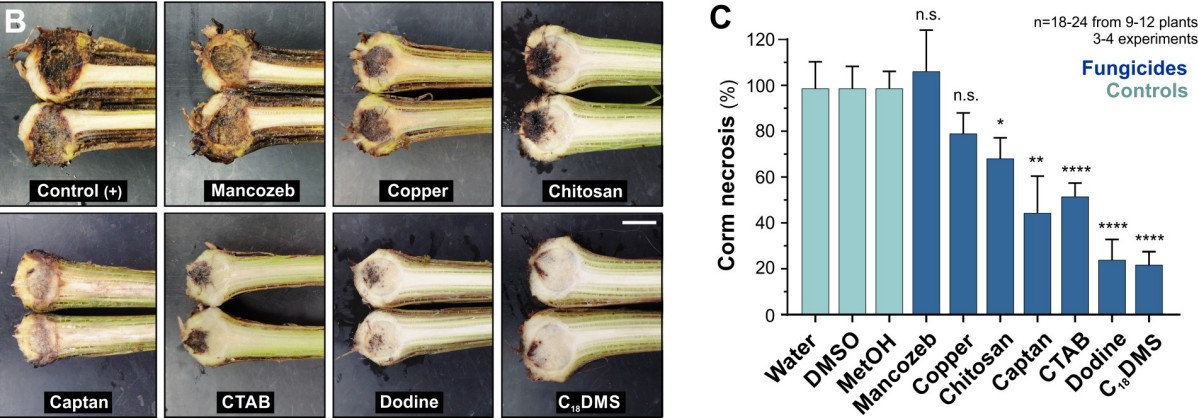

**Fig 7. Fungicides and protection of bananas against Panama disease. A** Whole-plant symptoms of Panama disease at 56 days after root inoculation with chlamydospores, followed by 2 applications of fungicides or the solvents 0.14% (v v$^{-1}$) DMSO or 0.16% (v v$^{-1}$) methanol in water (Control (+)). The negative control (Control (-)) was only treated with the solvent. **B** Corm necrosis in bananas at 56 days after root inoculation with chlamydospores, followed by 2 treatments with fungicides. Control (+) indicates inoculation with spores, followed by treatment with 0.16% (v v$^{-1}$) methanol (positive control). Scale bar = 2 cm. **C** Quantitative assessment of darkening of corm tissue after inoculation with chlamydospores followed by 2 treatments with fungicides. Banana corm necrosis was analysed 56 days after the first treatment. Light blue: Controls; dark blue: fungicide treatments. Bars in (**C**) show mean ± SEM from 18–24 measurements of 9–12 plants from 3–4 experiments; statistical comparison in (**C**) used Student's t-testing with Welch correction; n.s. = non-significant difference to respective control at two-tailed error probability of $P = 0.7074$ (Mancozeb) and $P = 0.1871$ (Copper); * = significant difference to control at two-tailed at $P = 0.0452$ (Chitosan); ** = significant difference to control at two-tailed at $P = 0.006$ (Captan); **** = significant difference to control at two-tailed $P<0.0001$. Plant infection assays used LMW chitosan, 200 µg ml$^{-1}$ (applied as 333 µg ml$^{-1}$ lactate salt); copper, 200 µg ml$^{-1}$ (applied as 786 µg ml$^{-1}$ copper (II) sulfate pentahydrate); mancozeb, 70 µg ml$^{-1}$; captan, 20 µg ml$^{-1}$; CTAB, 200 µg ml$^{-1}$; dodine, 200 µg ml$^{-1}$; C$_{18}$DMS, 200 µg ml$^{-1}$.

copper afforded no significant protection against Panama disease (Fig 7A–7C). Plants treated with LMW chitosan appeared healthier and showed slightly reduced corm disease symptoms (Fig 7B and 7C, corm necrosis reduced by 31.3%). This result conflicted with recent published data, showing ~85% reduction in disease symptoms in corms when 5-times higher concentrations of chitosan were applied to bare roots and in soil (1000 μg ml$^{-1}$, [20]). We therefore repeated our infection assays, using 3000 μg ml$^{-1}$ and 15000 μg ml$^{-1}$ LMW chitosan. These high concentrations did not fully suppress whole plant disease symptoms (S8A and S8B Fig). At 3000 μg ml$^{-1}$ LMW chitosan, we found >55% corm necrosis at (S8C Fig; statistically significantly different from control at two-tailed $P = 0.0038$). Moreover, corm symptoms were still apparent after treatment with 15000 μg ml$^{-1}$ LMW chitosan (S8D Fig). Thus, LMW chitosan does not fully protect bananas against Panama disease.

Captan showed attenuated leaf damage in plants (Fig 7A) and 55.1% reduced corm necrosis, as compared with the disease control (Fig 7B and 7C). However, of the 7 fungicides tested, captan was least reproducible in its protective activity. The best protection was endowed by treatment with MALCs, which significantly reduced corm necrosis (CTAB: 48.05%, dodine: 75.6%, C$_{18}$DMS: 77.3%, Fig 7B and 7C), and restricted yellowing to lower, older leaves (Fig 7A). A direct comparison of dodine- and C$_{18}$DMS-treated plants revealed that C$_{18}$DMS treatment resulted in the slightly healthier plants (Fig 8A). To quantify this phenotype, we developed a "plant health" assay that estimated the size of the aerial plant parts in digital images (see Methods). This protocol confirmed that C$_{18}$DMS-treatment resulted in improved plant health when compared with dodine (Fig 8B). Thus, MALCs have the potential to protect bananas against FocTR4-induced Panama disease.

### C$_{18}$DMS induces reactive oxygen species and apoptotic cell death in FocTR4

In *Z. tritici* dodine and C$_{18}$DMS depolarise mitochondria, thereby impairing ATP synthesis in the pathogen [58]. However, only C$_{18}$DMS induces ROS, thereby inducing apoptotic programmed cell death [58]. Next, we investigated if dodine and C$_{18}$DMS show this mode of action in FocTR4. We firstly stained chlamydospores with TMRM but found that this potential-sensing dye did not stain mitochondria in resting cells (Fig 9A–9C, stage I). After spore germination initiation in PDB, TMRM-positive mitochondria appear (Fig 9A–9C, stage II) and strong TMRM-staining was seen behind the growing germ tube tip (Fig 9A–9C, stage IV; Fig 9D). When germ tubes were incubated with dodine or C$_{18}$DMS, the TMRM signal was strongly reduced, suggesting that both MALCs depolarise mitochondria (Fig 9D). We applied DHR123 and FITC-VAD(OMe)-FMK staining and asked if mitochondrial depolarisation is accompanied by ROS development and apoptotic cell death. Indeed, we found that only C$_{18}$DMS induced ROS (Fig 9E) and apoptosis (Fig 9F) in FocTR4. Thus, C$_{18}$DMS, but not dodine, affects FocTR4 by (i) compromising mitochondrial respiration (ii) inducing ROS molecules (iii) initiating apoptotic programmed cell death.

## Discussion

### FocTR4 and FocR1 show innate resistance to single-site fungicides

Fungicides are considered ineffective against *Fusarium oxysporum* f. sp. *cubense* in the field [1,9] (www.promusa.org/Fungicides+used+in+banana +plantations). We tested the hypothesis that this is due to innate fungicide resistance and/or soil-related limitations. To address this, we developed new techniques to study this pathogen. Previous work assessed fungal colony diameter on agar plates as a measure of fungicide efficacy [13,16,17,18,20], which

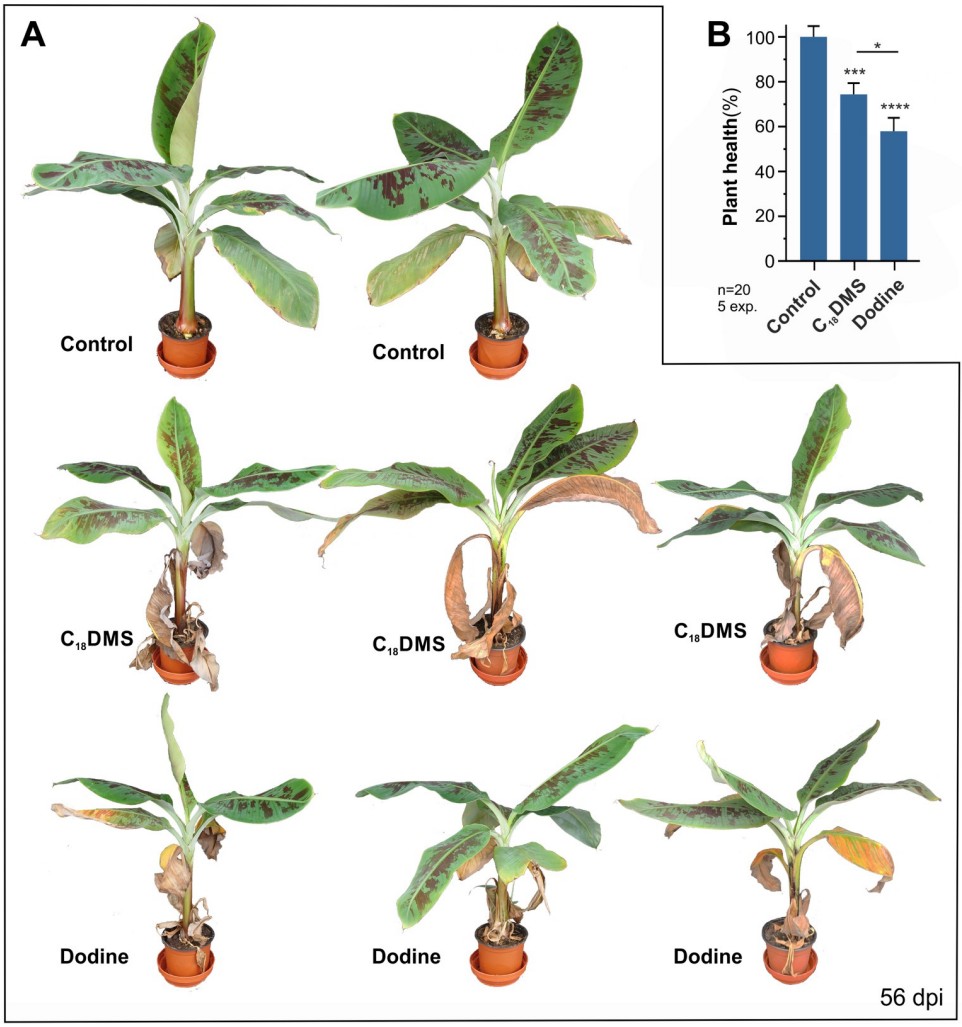

**Fig 8. Protective effect of C$_{18}$DMS and dodine in FocTR4-infected bananas. A** Cavendish banana plants, infected with FocTR4 chlamydospores, at 56 days after 2 treatments (day 0 and day 7) with C$_{18}$DMS, dodine or solvent control (0.16% MetOH (v v$^{-1}$)). **B** Bar chart showing plant health, assessed by measuring all aerial tissues in photographic images of 56 d-old plants, infected with chlamydospores and treated twice with C$_{18}$DMS or dodine. Control indicates healthy plants (100% plant health) that were not infected by FocTR4, but which were treated with the solvent methanol. Data in (**B**) are given as mean ± SEM; sample sizes indicated in graph; statistical comparison with control and between data sets used Student's t-testing with Welch correction; * = significant difference at two-tailed P = 0.0480; *** = significant difference at two-tailed P = 0.0007; **** = significant difference at two-tailed P<0.0001. Treatments were solvent control: 0.16% MetOH (v v$^{-1}$), dodine: 200 µg ml$^{-1}$ and C$_{18}$DMS: 200 µg ml$^{-1}$.

emphasises radial hyphal growth. However, we show that FocTR4 and FocR1 survive azole, SDHI and strobilurin treatment in PDA medium as swollen and irregular hyphae. These hyphae are short, and consequently, the colony does not form a peripheral "halo" of outward growing hyphae, but rather remains compact. Moreover, our quantitative 3 spore mortality analyses in SDW culture show that FocTR4 spores themselves are resistant to these fungicide classes. In addition, we report that spores survive benzimidazole treatment, including carbendazim, which has been used to control Fusarium wilt [19]. While carbendazim inhibited FocTR4 colony formation on agar, its inability to kill spores in liquid culture suggests that this, and other benzimidazoles, are fungistatic and not fungitoxic. Thus, FocTR4 carries remarkable

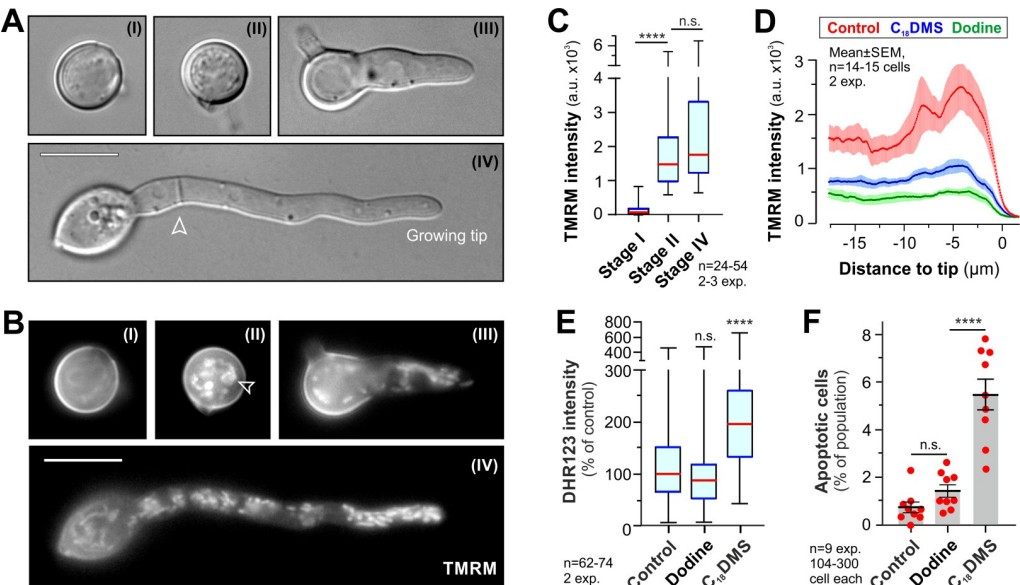

**Fig 9. The MoA of C$_{18}$DMS in germinating chlamydospores of FocTR4. A** Developmental stages of chlamydospore germination in potato dextrose medium. Dormant spores appear smooth (stage I); prior to germination they appear granular (stage II). Single germ tubes emerge, which form septa (arrowhead), extend at their growing tip and develop into hyphae (stage IV). Scale bar = 10. **B** Mitochondrial membrane potential, visualised by the cell-penetrating dye TMRM, in chlamydospores at different stages of germination. Note that almost no TMRM staining is seen in un-germinated spores (stage I), but mitochondria-associated fluorescence appear shortly before the germ tube emerges (stage II; arrowhead). Scale bar = 10 μm. **C** Mitochondrial membrane potential, indicated by TMRM fluorescence, in chlamydospores at stage (I) and (II) and in the germ tube (IV). **D** Intensity profile of TMRM fluorescence, indicative of mitochondrial membrane potential, along germ tubes of 7 h-old germlings of FocTR4, treated with the solvent (red curve), C$_{18}$DMS (blue curve) and dodine (green curve). Values are shown as mean ± SEM of 14–15 germlings. Note that both MALCs depolarises mitochondria. **E** Abundance of ROS, labelled with DHR123 in control, C$_{18}$DMS- and dodine-treated FocTR4 cells. C$_{18}$DMS induces ROS generation, whereas dodine reduces ROS. **F** Induction of cellular apoptosis, visualised by FITC-VAD(OMe)-FMK staining, in control, C$_{18}$DMS- and dodine-treated FocTR4 cells. C$_{18}$DMS alone induces apoptotic cell death. This activity was reported in *Z. tritici* [58]. Data in (**C, E**) are not normally distributed (Shapiro-Wilk test, P<0.05) and are shown as Whiskers' plots with 25/75 percentiles (blue line) and median (red line); data in (**D, F**) are given as mean ± SEM; red dots in (**F**) represent independent experiments; sample sizes indicated in all graphs; statistical comparison with control in (**F**) used Student's t-testing with Welch correction and in (**C, E**) non-parametric Mann-Whitney testing; n.s. = non-significant difference at two-tailed P = 0.0932 (**C**), P = 0.0993 (**E**) and two-tailed P = 0.1016 (**F**); **** = significant difference at two-tailed P<0.0001 (**C, E, F**). Treatments were 10 μg ml$^{-1}$, 30 min, for C$_{18}$DMS and dodine (**D, E**) and 10 μg ml$^{-1}$ of both compounds for 24 h (**F**). Control in (**D, E, F**) indicates the use of an equivalent volume of the solvent methanol.

innate resistance against 4 major classes of single-site fungicides, which helps it overcome certain chemical control strategies.

## Fungicide-tolerant persisters in FocTR4 show a pronounced transcriptional response

We report that a sub-population of FocTR4 and FocR1 cells, when incubated in nutrient-rich media, survive and grow as hyphae at high concentration of azoles, SDHIs and strobilurins. This demonstrates that hyphae can cope with very high fungicide loads. Moreover, water-incubated FocT4 microconidia, macroconidia and chlamydospores did not germinate, but still survived 10 day-treatment of 100 μg ml$^{-1}$ azoles, SDHIs and strobilurins. These spores are likely to be inactive in water. Thus, the mechanism underpinning their fungicide tolerance may differ from that found in persister hyphae. In human pathogenic fungi, fungicide persisters are classified as "fungicide tolerant" [62]. As we demonstrate that all morphotypes of FocTR4

(hyphae, microconidia, macroconidia and chlamydospores) are pathogenic in bananas, these fungicide-tolerant persisters may prove problematic as inoculum sources in the field.

Surprisingly, all single-target site fungicides were ineffective in FocTR4.

Whilst mutations in enzymatic targets of azoles or SDHIs are reported to confer resistance in fungal pathogens [36,41], FocTR4 persisters did not carry newly-acquired mutations in target genes or their promoters. Moreover, we report that the ability to cope with high fungicide load is not genetically inherited. Instead, persisters showed a strong transcriptional response to fungicides. This included extreme up-regulation of an Erg11-like azole target gene (~660-fold), not found in the fungicide susceptible *Z. tritici* strain IPO323, and high up-regulation of an aberrant succinate dehydrogenase subunit 3 allele. Moreover, non-target site mechanisms may contribute to FocTR4 fungicide tolerance. We see strong up-regulation of pathway genes, putative efflux proteins and detoxifying enzymes, shown to confer fungicide resistance in other fungal pathogens [39,45,48]. Of particular interest is a putative ABC transporter (UniProt ID: X0JJA9) with its expression induced in azole-, SDHI- and strobilurin-treated FocTR4 persisters and with homologues providing drug resistance in *Trichophyton rubrum* [51], *Candida albicans* [63] and *Aspergillus fumigatus* [64]. Thus, this pump may participate in fungicide tolerance in FocTR4 persisters. However, many other putative resistance-associated genes are also up-regulated in persisters, suggesting that tolerance is a consequence of several co-operating mechanisms. Most of these genes are strongly-induced by fungicides (Table 2), supporting the notion that transcriptional regulation is a vital mechanism underpinning FocTR4 fungicide tolerance. The transcription factors involved are unknown, but one gene (UniProt ID: X0JJ38) is consistently induced in azole, SDHI and strobilurin persisters. This suggests a general role for this putative regulator in transcriptional responses to fungicides. More work is needed to understand how FocTR4 achieves high tolerance to such fungicides.

## Soil application alters the efficacy of fungicides in controlling FocTR4

We developed robust quantitative virulence assays on soil-grown bananas to test fungicide efficacy. Previous studies have described virulence assays, using bare root dips [13,18,20], unknown inoculum densities [13], or high fungicide concentrations applied in excessive volumes of water [18], which likely washes out inoculum. Here, we inoculated with defined numbers of chlamydospores, avoided transplanting of plants, calibrated water amounts to avoid run-off, and included Silwet L-77 for even soil distribution. In addition, disease symptom development was assessed in an unbiased and quantitative way, using image analysis techniques. This contrasts with qualitative assessments, such as scoring disease using a visual corm severity index [20]. That these technical differences are relevant is illustrated by our results regarding the ability of LMW chitosan to protect against *Fusarium* wilt in bananas. It was reported that 1000 μg ml$^{-1}$ chitosan reduces disease symptoms by ~85% [20], whereas our study reveals only ~41% attenuation, even at a 3-times higher chitosan concentration. However, Widodo et al. (2021) applied the fungicides to bare roots prior to transplanting the tissue-cultured banana plants [20]. This procedure is not relevant for the field, and we believe our soil surface treatment protocol better mimics field application, thereby allowing a more reliable assessment of fungicide efficacy.

We assessed the 7 multi-site fungicides, proven effective *in vitro*, in virulence assays and found that neither mancozeb, nor copper afforded significant protection. Both fungicides show anti-fungal activity against *Fusarium* species *in vitro* [65,66] (this study), however, in greenhouse-grown tomato assays even 1000 μg ml$^{-1}$ copper-based fungicides proved ineffective against *Fusarium* wilt [67]. Moreover, while mancozeb has been reported to be effective against *Fusarium oxysporum* in soil [68], the concentrations used were >10-fold higher than

in our study. As mancozeb negatively impacts soil microflora [65,68] and is banned in regions of the world (https://chemtrust.org/news/mancozeb-ban-eu/), such intense applications in banana fields are deemed unsustainable.

But why does soil reduce fungicide efficacy? Soil is composed of minerals, organic matter and disparate microbial communities [69]. Such complexity makes it difficult to understand the behaviour of soil-applied fungicides. Indeed, the fate of fungicides in soil depends on their biophysical properties. This determines their mobility in soil [30,31,32], which may underpin their differential efficacy in soil and *in vitro* cultures. However, this remains speculative and merits future research.

### MALCs protect bananas against Panama disease

Of all tested fungicides, only captan and the 3 MALCs afforded significant protection against disease in growing bananas. Captan was previously reported to protect tomato plants from *Fusarium oxysporum* f. sp. *lycopersici* [70], and chickpeas from *Fusarium oxysporum* f. sp. *cicero* [71]. Thus, captan is a promising agent to control Panama disease. However, we found significant variability in its efficacy in preventing corm necrosis. While higher concentrations of captan could improve performance, captan, like other fungicides, is known to affect non-target soil microorganisms [72] —a risk that needs to be considered when using this fungicide in banana fields.

The best protection against Panama disease was provided by dodine and $C_{18}DMS$. Cationic surfactant molecules are well-known to have anti-fungal activity [73] and quaternary ammonium compounds, such as BAC and DDAC, have been used to clear contaminated agricultural equipment from FocTR4 [15,16]. Both compounds kill spores [13,14,15,16], but were previously tested at concentrations likely to cause phytotoxicity and environmental impacts if extended to field usage [74]. We show that dodine and $C_{18}DMS$ protect against Panama disease at much lower concentrations and with no phytotoxicity at 27˚C, the optimum mean temperature for banana production [28]. This is likely related to their primary mode of action (MoA) in mitochondrial respiration, which results in impaired ATP production [58]. However, only $C_{18}DMS$ induces formation of mitochondrial ROS in FocTR4, thus triggering apoptotic programmed cell death in FocTR4, as previously reported in *Z. tritici* [58]. As apoptosis is an irreversible "suicide" programme [75], this activity may underpin the higher efficacy of this MALC against Panama disease.

In summary, we show here that FocTR4 carries innate resistance to the major single-site fungicide classes. However, some multi-site fungicides, and, in particular the MALCs, offer hope in our quest to protect bananas from FocTR4.

## Materials and methods

### Biological material

**Fungi.** *Fusarium oxysporum* f. sp. *cubense* Tropical Race 4 (FocTR4; strain II5 NRRL#54006, isolated from *Musa* spp. in Indonesia) and *Fusarium oxysporum* f. sp. *cubense* Race 1 (FocR1; strain CR1.1.A, VCG0120, isolated from *Musa* spp. in Costa Rica) were kindly provided by Prof. Gert Kema (Wageningen University, The Netherlands). Cells were grown in PDB and stored in 50% (v/v) glycerol at -80˚C. *Zymoseptoria tritici* strain IPO323 was obtained from Centraalbureau voor Schimmelcultures, Utrecht, The Netherlands (cbs 115943). Cells were grown in YG (yeast extract, 10 g l$^{-1}$, glucose 30 g l$^{-1}$), diluted into 70% NSY (nutrient broth 8 g l$^{-1}$, sucrose 5 g l$^{-1}$, yeast extract 1 g l$^{-1}$) glycerol and stored at -80˚C. Molecular cloning used *Saccharomyces cerevisiae* strain DS94.

**Bacteria.** Plasmids were propagated in *Escherichia coli* strain DH5α. *Agrobacterium tumefaciens*-mediated transformation of FocTR4 used strain EHA105. The bacteria were grown in DYT (double strength yeast extract/tryptone medium) at 37˚C and 28˚C, respectively.

**Plants.** *Musa acuminata* cv. Grand Nairn banana plantlets in agar were imported from Canary Islands (Cultesa, Tenerife, Spain) and transplanted into 500 ml plastic pots (12 cm diameter), containing 400 g John Innes No. 2 soil (J. Arthur Bowers, Dungannon, UK). The plants were grown in a glasshouse on a long-day light cycle (16 h light: 8 h dark), using artificial light (SON-T 230V 600W bulbs; Philips, Amsterdam, The Netherlands) at 27 ± 1˚C during the day, reflecting the optimum mean temperature for banana production [28], and 24˚C ± 1˚C at night. The plants were watered daily and fertilised weekly with Yara Tera Kristalon Label fertiliser solution (Yara UK Ltd, Grimsby, UK) at a rate of 0.5 g l$^{-1}$.

## Production of FocTR4 morphotypes

Media used for production of chlamydospores, macrospores, microspores and hyphae were supplemented with PenStrep at a final concentration of 50 mg l$^{-1}$ (Sigma–Aldrich, Gillingham, UK).

**Microconidia.** 100 ml potato dextrose broth (PDB, Sigma Aldrich, Gillingham, UK) were inoculated from -80˚C glycerol stocks and grown at 26˚C, 150 rpm, for 7 days, filtered through two layers of Miracloth (Merck Millipore, Hertfordshire, UK), pelletted by centrifugation and washed 3-times with SDW, The purity was 100% (100±0.0%, n = 3 preparations, >100 cells each).

## Hyphae

Microconidia were incubated in yeast peptone dextrose broth (YPD) at 26˚C, 150 rpm, for 4 d. Cell cultures were filtered through Miracloth and hyphae, retained on the Miracloth, harvested by submersion of the Miracloth in SDW in a 50 ml Falcon tube (Fisher Scientific, UK). After hand shaking for 30 sec (to remove attached microspores and break up the hyphae), the filtrate was re-filtered through Miracloth and retained hyphae harvested by submersion of the Miracloth in SDW. Hyphae were used immediately after harvesting. At this 4 day time-point and following this protocol, the hyphae were countable as individual fragments, ranging in size. Hyphal preparations were 97.3±0.9% pure (n = 3).

**Chlamydospores.** Soil broth was prepared with 250g of John Innes No. 2 soil (J. Arthur Bowers, Westland Horticulture Ltd, Dungannon) in 1 litre distilled water and autoclaved for 30 mins at 121˚C. The broth was left to cool down and settle before the supernatant was filtered through 0.34 mm Whatman 3MM Chromatography filter paper (GE Healthcare Life Sciences, UK), re- autoclaved and subsequently supplemented with 0.5g/L dextrose by filter-sterilisation. For production of chlamydospores, ~10$^{8}$ microconidia per liter were inoculated into soil broth and grown at 26˚C, 120 rpm, for 14–28 days. Chlamydospores, attached to hyphae, were harvested by filtering through Miracloth, and suspended in 40 ml sterile distilled water. Chlamydospores were released from hyphae using a Soniprep 150 sonicator (Sanyo, Osaka, Japan) for 5 minutes, using a 19 mm titanium probe. Residual hyphal fragments were removed by Miracloth filtration. Finally, chlamydospores were pelletted by centrifugation and washed 3-times with sterile distilled water. The purity of chlamydospore preparations was 95.5 ±1.4% (n = 3).

**Macroconidia.** Microconidia were plated on yeast peptone dextrose agar (bacteriological peptone, 20 g l$^{-1}$; glucose, 20 g l$^{-1}$; agar, 15 g l$^{-1}$) for 10d at 26˚C under constant illumination (Certomat BS-1 growth cabinet, Satorius, Göttingen, Germany with OSRAM (L18W/77 bulbs, 72 µmol m$^{-2}$ s$^{-1}$ at plate; https://www.osram.com). Spores were harvested into 5 ml SDW, filtered through Miracloth, pelletted by centrifugation and washed 3-times with sterile distilled water. The purity of macroconidia preparations was 94.2±1.9% (n = 3).

## Microscopy

**Transmission electron microscopy.** Fungal cells, prepared as described, were fixed in 2% (v v$^{-1}$) glutaraldehyde and 2% (v v$^{-1}$) paraformaldehyde in 0.1 M sodium cacodylate (pH 6.8), and post-fixed in 2% (w v$^{-1}$) potassium permanganate. After dehydration in an ethanol gradient (30%-100%), cells were embedded in Durcupan resin. 60 nm sections were contrasted in lead citrate and imaged using a JEM 1400 transmission electron microscope (JEOL Ltd., Tokyo, Japan), fitted with a digital camera (ES 1000W, Gatan, Pleasanton, USA) at the Exeter Bioimaging Centre.

(https://biosciences.exeter.ac.uk/bioimaging/).

Scanning electron microscopy. Cultures grown on potato dextrose agar at 25˚C for 2 days were carefully excised, using a razor blade, and immersed in fixative (2% (v v$^{-1}$) glutaraldehyde, 2% (v v$^{-1}$) paraformaldehyde in 0.2M PIPES (Sigma-Aldrich, pH 7.2) for 2 h at room temperature. Samples were washed 3 times for 5 min in PIPES buffer before post-fixation in 1% aqueous osmium tetroxide (w v$^{-1}$) for 1 h, then washed 3 times for 5 min in deionized water. Cells were then dehydrated using a graded ethanol series (from 30% up to 100% ethanol) and finally incubated for 3 min in HMDS (hexamethyldisilazane, Merck, Gillingham, UK) before air drying and mounting on aluminium stubs. Samples were coated with 10 nm gold-palladium and imaged using a Zeiss GeminiSEM 500 operated at 1.5 kV, with a SE2 detector at the Exeter Bioimaging Centre (https://biosciences.exeter.ac.uk/bioimaging/).

Laser-based epifluorescent microscopy was performed as described [58]. In brief, cells were placed on a 2% (v v$^{-1}$) agar cushion and observed using a motorized inverted microscope (IX81TIRF/IX83; Olympus, Hamburg, Germany) and either a PlanApo 100×/1.45 Oil TIRF objective, an UPlanSApo 60x/1.35 Oil objective or an UApoN340 40x /1.35 Oil objective (Olympus). Fluorescent probes and proteins were excited using a VS-LMS4 Laser Merge System (488 nm at 75 mW and 561 nm at 75 mW, Visitron Systems, Puchheim, Germany). Images were captured using a CoolSNAP HQ2 camera (Photometrics, Tucson, USA). All parts were controlled by software package VisiView (Visitron Systems). General bright field DIC microscopy used built-in illumination. Images were analysed using the software package Meta-Morph 7.8x (Molecular Devices, Wokingham, UK).

*In planta* confocal microscopy was carried out as described [26] on a Leica TCS SP8 laser scanning confocal microscope (Leica, Wetzlar, Germany). To visualise growth of FocTR4 on/in roots, 4-week-old plants in 12cm pots were inoculated with 1x10$^5$ chlamydospores per g of soil. For observation, plants were uprooted and soil was washed from the root ball by rinsing under running water. Tertiary roots were cut from the root ball and mounted on microscope slides on pads of Carolina observation gel (Burlington, NC, USA) and observed using a HC PL APO CS2 63x/1.40 oil immersion lens. Samples were excited using 488 nm and a 561 nm lasers. HyD detectors were set at 518nm - 551nm for GFP, and 640nm - 668nm for root tissue autofluorescence.

## Cell staining methods

Staining of lipid droplets in chlamydospores was by incubating cells for 10 mins, at room temperature (RT), in 5 μl ml$^{-1}$ BODIPY 493/503 (Thermo Fisher Scientific, Loughborough, UK stock solution 1 mg ml$^{-1}$ in DMSO). Cell walls were co-stained in 5 μl ml$^{-1}$ Calcofluor White (Sigma, stock solution 1 mg ml$^{-1}$ in phosphate buffer, pH 7).

Mitochondrial membrane potential was visualised using the Image-iT TMRM reagent (Thermo Fisher Scientific). Pre-cultures of strains FocTR4_eGFP-Sso1 or IPO323_eGFP-Sso1 [76] were grown in PDB for 24 h at 25˚C, at 200 rpm, or in YG for 24 h, 18˚C, at 150 rpm. One ml of of these pre-cultures was incubated with 500 μg ml$^{-1}$ azoxystrobin or corresponding

amounts of the solvent methanol alone for 24 h at room temperature in the dark on a SB2 Rotator (Bibby Scientific Limited, Stone, UK; details of incubation time and concentrations are given in the figure legends). Subsequently, 1 μl of TMRM was added to 1 ml of cell suspension, followed by additional 10 min incubation in the dark and immediate microscopic examination. Images were analysed in MetaMorph and intensity values in the cell corrected for the culture background. To detect the mitochondrial potential in germlings, chlamydospores were incubated in PDB at 25˚C, 150 rpm, for 2–7 h. One ml of the spore suspension was treated for 30 min with $C_{18}$DMS or dodine at 10 μg ml$^{-1}$ or corresponding amounts of the solvent methanol (control) for 30 min at room temperature, in the dark, on a SB2 Rotator. The fluorescent intensity of the TMRM was determined as decribed above. Intensity profiles TMRM fluorescence in germ tubes was done using the "line-scan" function in MetaMorph.

Reactive oxygen species detection was as descibed [58]. Briefly, *Z. tritici* IPO323 or FocTR4 cells were incubated for 24 h with 500 μg ml$^{-1}$ azoxystrobin, or the solvent methanol, followed by staining with dihydrorhodamine 123 (DHR123, Sigma-Aldrich, Gillingham, UK). One μl DHR123 was added to 1 ml cell suspension), followed by 15 min incubation at room temperature in the dark on a SB2 Rotator. Cells were imaged and the average intensity of whole cells was measured in digital images. All values were corrected for the corresponding image background. For ROS detection in chlamydospores, spores were pre-incubated in PDB for 2 h, followed by 24 h incubation in PDB supplemented with either 10 μg ml$^{-1}$ $C_{18}$DMS, 10 μg ml$^{-1}$ dodine or the corresponding amount of the solvent methanol. ROS was detected using DHR123 as decribed above.

Metacaspase activity in early apoptotic cells was visualised using the CaspACE FITC-VAD (OMe)-FMK In Situ Marker assay (Promega, Madison, USA), as descibed [58]. *Z. tritici* or FocTR4 cells were treated with 500 μg ml$^{-1}$ azoxystrobin, or the solvent methanol for 24 h. 100 μl of treated cell suspension was incubated with a mixture of 0.1 μl of FITC-VAD(OMe)-FMK and 0.1 μl propidium iodide (Sigma Aldrich; stock: 1 mg ml$^{-1}$ in ddH$_2$O) for 1 h at RT, in the dark, on a SB2 Rotator. Cells were sedimented by centrifugation at 5000 rpm for 5 min, and washed with 100 μl fresh PDB medium. Cells were co-imaged with a 488 nm and a 561 nm laser; only cells that showed green- but no red-fluorescence were considered apoptotic. Metacaspase activity in germinating chlamydospores was detected by pre-incubated of spores in PDB for 2 h, followed by 24 h incubation in PDB supplemented with 10 μg ml$^{-1}$ $C_{18}$DMS, 10 μg ml$^{-1}$ dodine or the corresponding amount of the solvent methanol. FITC-VAD(OMe)-FMK staining was as decribed above.

## FocTR4 and FocR1 colony growth in response to fungicide treatment

All fungicides were obtained from Sigma-Aldrich, with the exception of cetrimonium bromide (CTAB; Thermo Fisher Scientific) and $C_{18}$DMS, which was synthesised by Dr. Mark Wood (University of Exeter, UK). Chitosan oligosaccharide lactate and copper(II) sulfate pentahydrate stock solutions were prepared in water, whilst carbdendazim, mancozeb, chlorothalonil and captan stock solutions were in dimethyl sulfoxide (DMSO). All other fungicide stock solutions were in methanol (Thermo Fisher Scientific). YPD agar plates (for growth of *Z. tritici* IPO323) and potato dextrose agar plates (PDA: for growth of FocTR4and FocR1) were supplemented with fungicides or their respective solvents (100% control). FocTR4 and FocR1 microconidia suspensions were diluted (concentrations were 10$^6$ cells ml$^{-1}$, 10$^5$ ml$^{-1}$, 10$^4$ ml$^{-1}$, 10$^3$ ml$^{-1}$; here and in the following cells were counted using a Cellometer Auto 1000 cell counter (Nexcelom Biosciences, Lawrence, USA). Drops of 5 μl cell suspension were placed on the agar plates, followed by incubation for 2 days at at 25˚C. For *Z. tritici* assays, diluted cell suspension were prepared (concentrations were 10$^6$ cells ml$^{-1}$, 0.5 x 10$^6$ ml$^{-1}$, 0.25 x 10$^6$ ml$^{-1}$, 1.25 x 10$^5$,

$6.25 \times 10^4$). Drops of 5 μl were placed on the agar plates, followed by incubation for 5 days at at 18˚C.

Digital images of plates were obtained using an Epson Perfection V750 Pro scanner (Epson, Hemel Hempstead, UK). Growth inhibition was assessed by measuring the integrated intensity of the third highest dilution for IPO323 and second dilution for FocTR4 and FocR1. Digital images were converted to grey-scale using Photoshop CS6 (Adobe Inc., San Jose, USA), an area of interest was drawn around the colony and the integrated intensity of all pixels in this area determined. Next, the area of interest was moved to a fungal cell-free region and the integrated intensity of this background was measured. After correcting for this background, the values for colony formation on control plates, containing the solvents only, were set to 100% and all other measurements were compared. The resulting values represented the relative cell density (= colony brightness) and were plotted and analysed in GraphPad Prism 6 to obtain fungicide dose-response curves.

### Test for inherited fungicide tolerance in FocTR4 using plate growth assays

FocTR4 microconidia were grown in 20 ml PDB, supplemented with 3 μg ml$^{-1}$ epoxiconazole, 500 μg ml$^{-1}$ azoxystrobin or 500 μg ml$^{-1}$ fluxapyroxad, at 25˚C for 2 days. For control experiments, microconidia were grown in the presence of 2.5% (v v$^{-1}$) methanol. This was followed by filtering through Miracloth, which removed hyphae. Microcondia, grown in the presence of fungicide (= pre-treated) or the solvent only (= untreated) were plated onto fungixcide-containing potato dextrose agar plates and incubated for 2 days at 25˚C. Plates were scanned and growth inhibition quantified (as described above).

### Fungicide toxicity to FocTR4 in in liquid culture assays

FocTR4 macroconidia, microconidia and chlamydospores were prepared as described above and diluted in SDW. The cell density was adjusted to $5 \times 10^6$ ml$^{-1}$ and fungicides were added to a final concentration of 100 μg ml$^{-1}$ (azoxystrobin, pyraclostrobin, trifloxystrobin, bixafen, boscalid, fluxapyroxad, triticonazole, tebuconazole, epoxiconazole, LMW chitosan [applied as 167 μg ml$^{-1}$ lactate salt], copper [applied as 393 μg ml$^{-1}$ copper(II) sulfate pentahydrate], garlic oil, thiophanate methyl, tebuconazole, CTAB, dodine, $C_{18}$DMS). Highly efficient category I fungicides were applied at 5-times their minimal inhibitory concentration (MIC; (carbendazim 4.50 μg ml$^{-1}$, captan 10 μg ml$^{-1}$, mancozeb 35 μg ml$^{-1}$, chlorothanonil 5 μg ml$^{-1}$). The suspensions were incubated on in the dark on a rotation wheel at 85 rpm at 25˚C. Cells were harvested after 10 days by centrifugation and stained with the red-fluorescent LIVE/DEAD Fixable Red Dead Cell Stain Kit (Molecular Probes Thermofisher, UK), used at a final concentration of 1 μl ml$^{-1}$. Cell death was assessed in microscopic images, taken with wide-field and red-fluorescence settings at the Olympus inverted microscope. Cells were considered dead if red-fluorescence was found within the borders of the cell. To assess "persister" survival of FocTR4 and IPO323 in high concentration of fungicides (3 μg ml$^{-1}$ epoxiconazole, 500 μg ml$^{-1}$ azoxystrobin or 500 μg ml$^{-1}$ fluxapyroxad), pre-cultures were grown for 2 days at 25˚C or 18˚C, 150 rpm and 5 ml of this cell suspension was used to inoculate fresh PDB medium, supplemented with identical concentrations of fungicides. Samples were taken after 2 days, 5 days and 9 days and mortality assessed using LIVE/DEAD staining.

### Transcriptome sequencing in the 4 morphotypes

Total RNA, including nuclear and mitochondrial transcripts, was prepared from the four different FocTR4 morphotypes (see above), after grinding cells in liquid nitrogen, using the RNeasy plant mini kit (Qiagen, Manchester, UK), following manufacturer's instructions.

Genomic DNA was removed from RNA samples by treating with DNase I (Qiagen) at 28˚C for 15 min. Transcriptome library preparation, and sequencing of 3–6 biological replicates per condition, was undertaken by the Exeter Sequencing Service (University of Exeter, UK). RNA concentration was quantified using Qubit RNA Assay kit and a Qubit 2.0 Fluorometer (Thermo Fisher Scientific). RNA integrity was assessed using the Tapestation 4200 automated electrophoresis system (Agilent Technologies, Waldbronn, Germany). Nuclear transcriptome libraries were created from 500ng total RNA using TruSeq Stranded mRNA Library Prep Kit (Illumina, San Diego, USA) and mRNA was isolated using poly-A oligo-attached magnetic beads prior to sequencing as 125 or 150 base-pair paired-end reads on either the HiSeq 2500 or NovaSeq platform (Illumina). Mitochondrial transcriptome libraries were created from 100ng total RNA. The ribosomal RNA was depleted using pan-fungi ribopool (siTOOLs Bio-tech, Planegg, Germany) and the libraries were generated using the Truseq stranded mRNA library preparation kit (Illumina), according to the manufacturer's protocols. 150 bp base-pair paired-end reads were sequenced on NovaSeq platform (Illumina). Adaptor and base quality trimming of raw FASTQ data used Cutadapt v1.13 (minimum overlap = 5, base quality = 22, length = 63) or FastP v 0.20.1 (min read length = 75). Read quality was assessed as described previously. Variability between transcriptome data sets was determined by principal component analysis using DESeq2 v1.14.1 (web addresses and references for all programs provided in S1 Table).

## Functional annotation of proteins in fungicide-treated FocT4 and IPO323

Functional annotation was undertaken using BLASTp with the non-redundant protein database (v2.2.24 for FocTR4, v2.12.0 for IPO323) and Interproscan (v5.32–71.0 for FocTR4 and v5.52–86.0 for IPO323) with GO term and pathways retrieval (S1 Table). Putative resisance associated candidate genes were screened in the derived Excel tables, using domain names of enzymes, implied in (i) metabolic detoxification (including cytochrome P450 monooxygenases, dehydrogenases, oxidases, esterases, reductases, glucuronosyltransferases, sulfotransferases, glutathione S-transferases [49,50], (ii) membrane transporters that may function as fungicide efflux pumps (ABC and major facilitator superfamily transporters [77]. Known fungicide-target proteins and enzyme of the fungicide-targeted pathway, as well as other proteins of interest, were identified using BLASTp searches with *S. cerevisiae* or human homologues, received from NCBI, in the FocTR4 database at EnsemblFungi or the *Z. tritici* IPO323 database at FungiDB. FocTR4 and IPO323 protein sequences were obtained from UniProt (S1 Table). All hits were confirmed by reverse BLAST searches at NCBI. Putative ABC and MFS transporters were analysed in the Transporter Classification Database TCDB. Protein domains were predicted using InterProScan, secretion signal peptides were determined in SignalP-5.0, mitochondrial targeting peptides were predicted in TargetP-2.0, mitochondrial sub-location was predicted using DeepMito and protein cysteine-contents was calculated using the Quest Calculate Peptide and Protein Molecular Weight Calculator. Sequence comparisons were done using EMBOSS Needle or Clustal Omega. The impact of residue substitutions was estimated using the SIFT server. The Erg11-homologue heat map was generated in Excel (Microsoft).

## Sequence data alignment and differential expression

Nuclear RNA reads were aligned to the FocTR4 II5 reference genome (FO_II5_V1/ GCA_000260195) [37] or IPO323 reference genome (MG2/GCA_000219625) [78] and mitochondrial RNA reads were aligned to the FocTR4 II5 mitochondrial reference genome LT906347.1 [38], and IPO323 to the equivalent mitochondrial reference genome (NC_010222.1, [79]), both using TopHat2 v2.1. Read counts for each gene were generated

using HTSeq-count v0.10.0 prior to normalization and identification of differentially expressed genes (false detection rate p-adj<0.05) using DESeq2 v1.14.1.

## Molecular cloning

Vector pHeGFPSso1 was generated by *in vivo* recombination in *S. cerevisiae* DS94 (MATα, *ura3-52*, *trp1-1*, *leu2-3*, *his3-111* and *lys2-801*) following published procedures [80]. DNA fragments were transformed into *S. cerevisiae* for *in vivo* recombination, and plasmids amplified in *E. coli* DH5α. All restriction enzymes and reagents were obtained from New England Biolabs (Herts, UK).

The vector pHeGFPSso1, designed for random integration into the genome, contains the gene for enhanced green-fluorescent protein fused to the 5-prime end of the FocTR4 *sso1* gene (UniProt ID: N1RDI3), flanked by the promoter and terminator of FocTR4 alpha-tubulin (UniProt ID: N1R6Q4). The DNA fragment, prepared for *in vivo* recombination in *S. cerevisiae* were: (i) 9760 bp fragment was derived from vector pCGEN-YR [81], obtained by digestion with restriction enzymes *Xba*I and *Zra*I; (ii) 1151 bp frament, containing the alpha-tubulin promoter, amplified from genomic DNA of FocTR4, using primers SK-Fox-5 and SK-Fox-6 (S9 Table for all primers); (iii) 717 bp carrying the gene for enhanced green-fluorescent protein, but no stop codon, amplified with primers SK-Sep-16 and SK-Sep-17 from vector pCeGFP [80]; (iv) 1100 bp frament, encoding the open reading frame of FoTR4 *sso1*, amplified using SK-Fox-11 and SK-Fox-12 from FocTR4 genomic DNA; (v) 1023 bp frament of DNA, carrying the alpha-tubulin terminator, amplified with SK-Fox-13 and SK-Fox-8 from FocTR4 genomic DNA; (vi) 1806 bp, encoding the hygromycin resistance cassette, amplified with primers SK-Fox-16 and SK-Fox-17 from plasmid pCHYG [81].

## FocTR4 transformation

*A. tumefaciens*-mediated transformation of FocTR4 was performed, modified from procedures described for *Z. tritici* [80]. The vector pHeGFPSso1 was transformed into *A. tumefaciens* strain EHA105 by heat shock and transformants selected on DYT agar medium, supplemented with 20 μg ml$^{-1}$ rifampicin and 50 μg ml$^{-1}$ kanamycin (Sigma–Aldrich). The transformants were confirmed by colony PCR using primers SK-Sep-16 and Fox-12 (for vector pHeGFPSso1), grown in 10 ml DYT medium supplemented with 20 μg ml$^{-1}$ rifampicin (Melford, Ipswich, UK) and 50 μg ml$^{-1}$ kanamycin 28˚C with 200 rpm. The overnight cultures were diluted to an optical density of 0.15 at 660 nm in *Agrobacterium* induction medium AIM [80], supplemented with 200 μM acetosyringone (Sigma–Aldrich) and grown at 28˚C, at 200 rpm, until an optical density reached to 0.3–0.35 (4–5 hours). The *A. tumefaciens* cultures which contain the desired vectors were mixed with an equal volume of FocTR4 microconidia (1x10$^5$ spores ml$^{-1}$) and 200 μl of *A. tumefaciens*–FocTR4 plated onto nitrocellulose filters (AA packaging limited, Preston, UK) placed on AIM agar plates supplemented with 200 μM acetosyringone and grown at 25˚C for 2 days. Nitrocellulose filters were transferred onto PDA plates (Oxoid, Basingstoke, UK) containing 100 μg ml$^{-1}$ cefotaxime (Melford), 100 μg ml$^{-1}$ timentin (Melford) and 100 μg ml$^{-1}$ hygromycin B (Invivogen, Toulouse, France) and incubated at 25˚C for 5–7 days. Individual colonies were transferred onto PDA plates, containing 100 μg ml$^{-1}$ cefotaxime, 100 μg ml$^{-1}$ timentin and 100 μg ml$^{-1}$ hygromycin B, and grown at 25˚C for 3–4 days.

## FocTR4 spore forms and hyphal virulence assays on bananas

Virulence assays were undertaken by root drench treatment of 8 to 12 week-old banana plants, grown at 27 ± 1˚C, which carried 4–5 fully formed leaves. Briefly, the root system was

wounded by cutting once through the entire soil area, at a 45-degree angle, with a garden trowel (3/8 gauging trowel, Marshalltown Tools, Braintree, Essex, UK). 50 ml of cell suspensions, containing microconidia, macroconidia, chlamydospores or hyphae were added to the soil, resulting in $1 \times 10^5$ propagules per gram of soil. The inoculated plants were grown for 56 days in the greenhouse, as described above. Corm necrosis, an internal marker of disease severity, and the health of the entire plant was assessed from digital images, taken under standardised conditions. To quantify corm necrosis, images of cut stems were converted to grey-scale and contrast-inverted, using MetaMorph, which converted dark necrosis areas into bright signals. In each image, the average pixel signal intensity in both halves of the corm and in a defined background area was measured. Image variations were normalised by adjusting all background signal intensities, providing corrected values for signal intensities in corms. The values of non-infected plants (negative control) were subtracted from all measurements. The values for infected plants that were treated with water, DMSO or methanol (infected "solvent" only controls, as described above) were set to 100% (= maximal symptoms) and measurements related to their respective treatments (copper and LMW chitosan to water control; mancozeb to DMSO control; all other compounds to methanol control). Whole plant health was quantified by subtracting the dark grey-coloured background in colour-images of entire banana plants and by removing dead leaves from the images, using the Magic Wand Tool and manual adjustment in Photoshop CS6, Version 13.0.1. Images were converted to grey scale, and imported into MetaMorph, were the area of all visible banana leaves and stem was automatically determined, using the auto-threshold and automated region creation functions. The area covered by control plants was set to 100% "plant health" and areas of fungicide-treated plants were compared against this value. Data analysis and presentation used Excel and Prism6 or Prism9.

## Fungicide efficacy against FocTR4 infected banana plants

Eight to 12 week-old banana transplant plants, grown in 400 g soil in 500 ml pots, were wounded, as described above. Chlamydospore suspension was poured onto the soil surface at a final concentration of $10^5$ spores per gram of soil. After 1 h of settlement, 50 ml of fungicide solution was added to the top of the soil. Concentrations were: 200 μg ml$^{-1}$ for CTAB, LMW chitosan [applied as 333 lactate salt] (or at 3000 or 15000 μg ml$^{-1}$ in additonal experiments), copper [applied as 786 μg ml-1 copper(II) sulfate pentahydrate], dodine, $C_{18}$DMS. Mancozeb was used at 70 μg ml$^{-1}$ and captan at 20 μg ml$^{-1}$. Negative controls were either methanol or DMSO, applied at concentrations equal to when used as solvent in the chemistries above. Treatments were repeated after 7 d. All solutions were supplemented with 0.1% (v v$^{-1}$) Silwet L-77, which ensured even penetration of the fungicide solutions in the soil. Plants were assessed after 56 d.

## Fungicide phytotoxicity assays on leaves and whole plant

Leaf droplet assays were carried out on the most recently fully-formed leaves of 8-week-old plants, grown at 27 ± 1°C, which reflects the optimal temperature for banana production [28]. 10 μl droplets of fungicide solutions were applied adaxially and incubated for 24 h at 27°C. Concentrations and controls were as described above, including BAC at 50 mg ml$^{-1}$ as a positive control. Images were taken after 24 h. Root drench assays were carried out on 8 week-old plants. 50 ml of fungicide solution, or BAC as the positive control, were added to the plant pots twice, on day 0 and on day 7, and plants were grown at 27°C /24°C and symptoms assessed on day 21. All concentrations were as in leaf droplet assays. Leaf droplet and root drench assays were carried out under banana plant growth conditions described above.

## Searching for fungicide target site mutations

FocTR4 microconidia were grown in 20 ml PDB, supplemented with 3 µg ml$^{-1}$ epoxiconazole, 500 µg ml$^{-1}$ azoxystrobin, or 500 µg ml$^{-1}$ fluxapyroxad, at 25˚C for 2 days. The genomic DNA (nuclear and mitochondrial) was isolated; genomic DNA library preparations and sequencing were undertaken by the Exeter Sequencing Service (University of Exeter, UK). Libraries were created using the Illumina NEBNext protocol and sequenced as 150 base-pair, paired end reads on an Illumina NovaSeq 6000 SP flowcell. Adaptor and base quality trimming of raw FASTQ data used FastP v0.20.1 (min read length = 75; S1 Table). Raw and trimmed read quality was assessed with FastQC v0.11.4 and sample composition assessed for contaminants using FastQScreen v0.5.2. High quality reads were aligned to the nuclear reference genome FocTR4 54006 (GCA_000260195) and mitochondrial reference genome (LT906347.1), using bwa mem (S1 Table). Picard Tools (Broad institute), SamFormatConverter, FixMateInformation and MarkDuplicates were used to convert from SAM to BAM format, fix discrepant mate pair information and mark PCR duplicate reads, respectively. GATK RealignerTargetCreator and IndelRealigner were used to locally realign highly variable regions to produce the BAM files. Target genes were manually inspected for variants using IGV.

## Supporting information

**S1 Fig. The effect of fungicides on growth of *Z. tritici* strain IPO323 on solid medium.**
(PDF)

**S2 Fig. Two phase response of FocTR4 colony formation to fluxapyroxad.**
(PDF)

**S3 Fig. Mortality of FocTR4 and *Z. tritici* IPO323 in liquid cultures (PDB) and appearance of persister hyphae.**
(PDF)

**S4 Fig. Expression of TCA cycle genes in fluxapyroxad-treated FocTR4 and IPO323 cells.**
(PDF)

**S5 Fig. Expression of genes of the ergosterol biosynthetic pathways in epoxiconazole-treated FocTR4 and IPO323 cells.**
(PDF)

**S6 Fig. Cell wall thickness in FocTR4 morphotypes.**
(PDF)

**S7 Fig. Phytotoxicity of fungicide solutions in bananas.**
(PDF)

**S8 Fig. Panama disease symptoms in infected chitosan-treated plants.**
(PDF)

**S1 Table. Bioinformatic tools used in this study.**
(PDF)

**S2 Table. Inhibition of plate growth of IPO323.**
(PDF)

**S3 Table. Putative fungicide targets in FocTR4.**
(PDF)

**S4 Table. Up-regulated genes in epoxiconazole-, fluxapyroxad- and azoxystrobin-treated FocTR4 cells.**
(XLSX)

**S5 Table. Up-regulated genes in epoxiconazole-, fluxapyroxad- and azoxystrobin-treated IPO323 cells.**
(XLSX)

**S6 Table. Comparison of putative fungicide tolerance-related genes in FocTR4 and IPO323.**
(XLSX)

**S7 Table. Expression of mitochondrial genes in azoxystrobin-treated FocTR4 and IPO323.**
(XLSX)

**S8 Table. Expression of proteins with unknown function in azoxystrobin-treated FocTR4 and IPO323.**
(XLSX)

**S9 Table. Sequences of cloning primers.**
(PDF)

**S1 Data. Numerical data and statistical analysis.**
(XLSX)

**S1 Video. 3D-animated PCA plot of the transcriptome of 4 FocTR4 morphotypes Each dot represents a single RNA preparation.**
(GIF)

## Acknowledgments

The authors are grateful to Connor Simpson for his dedicated technical support. We thank Weibin Ma and Ian Leaves, for help with fungicide plate assays and RNA preparations. Dr. Christian Hacker, Bioimaging Centre Exeter, is acknowledged for performing the electron microscopy work and Dr. Karen Moore, Exeter Sequencing Service, for help with sequencing the mitochondrial transcriptome. We also wish to thank Dr. Mark Wood, Exeter, UK, for synthesis of $C_{18}DMS$; Dr Harold Meijer for his help and advice on FocTR4 infection assays and, finally, we are grateful to Prof. Gert Kema, who provided the FocTR4 and FocR1 strains, and who inspired this study. Prof. Sarah Jane Gurr is a Fellow in the "Fungal Kingdom: Opportunities and Threats" programme of the Canadian Institute for Advanced Research (CIFAR).

## Author Contributions

**Conceptualization:** Gero Steinberg.

**Data curation:** Stuart Cannon.

**Formal analysis:** William Kay, Martin Schuster, Gero Steinberg.

**Funding acquisition:** Sarah Jane Gurr, Gero Steinberg.

**Investigation:** William Kay, Martin Schuster, Gero Steinberg.

**Methodology:** Stuart Cannon, William Kay, Sreedhar Kilaru, Sarah Jane Gurr, Gero Steinberg.

**Project administration:** Sarah Jane Gurr, Gero Steinberg.

**Resources:** Sreedhar Kilaru.

**Software:** Stuart Cannon.

**Supervision:** Sarah Jane Gurr, Gero Steinberg.

**Validation:** Gero Steinberg.

**Visualization:** William Kay, Martin Schuster, Gero Steinberg.

**Writing – original draft:** Sarah Jane Gurr, Gero Steinberg.

**Writing – review & editing:** Stuart Cannon, William Kay, Sreedhar Kilaru, Martin Schuster, Sarah Jane Gurr, Gero Steinberg.

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
