## [Decision Letter · Decision Letter 0]

25 Jun 2022

Dear Prof. Steinberg,

Thank you very much for submitting your manuscript "Multi-site fungicides suppress banana Panama disease, caused by Fusarium oxysporum f. sp. cubense Tropical Race 4" for consideration at PLOS Pathogens. As with all papers reviewed by the journal, your manuscript was reviewed by members of the editorial board and by several independent reviewers. In light of the reviews (below this email), we would like to invite the resubmission of a significantly-revised version that takes into account the reviewers' comments.

We cannot make any decision about publication until we have seen the revised manuscript and your response to the reviewers' comments. Your revised manuscript is also likely to be sent to reviewers for further evaluation.

Sincerely,

Bart Thomma

Handling and Section Editor

PLOS Pathogens

Kasturi Haldar

Editor-in-Chief

PLOS Pathogens

orcid.org/0000-0001-5065-158X

Michael Malim

Editor-in-Chief

PLOS Pathogens

orcid.org/0000-0002-7699-2064

Reviewer's Responses to Questions

**Part I - Summary**

Reviewer #1: In this study, the authors present a quite diverse collection of data about growth, spore types, spore-type-specific transcriptomes, infection and fungicide resistance of the banana pathogen FocTR4. The central conclusions of this study are i) FocTR4 has innate resistance of against a wide range of site-specific fungicides, ii) this innate resistance is due not to target resistance but to high upregulation of genes conferring resistance in a subpopulation of so-called persister cells, and iii) FocTR4 is sensitive only against two multisite fungicides, and this sensitivity offers a potential strategy for chemical control. While the last observation could have applied value, scientifically most interesting is the proposed new mechanism of innate fungicide resistance, due to the survival of special ‘aberrant hyphae’ which show strong upregulation of fungicide target and drug efflux transprorters.

This study contains interesting data about a devastating banana pathogen for which disease control solutions are urgently needed. However, there are several weaknesses in the manuscript that needs to be adressed before publication is recommended. First, as the focus is on the unusual fungicide resistance behaviour of FocTR4, the comparison with an unrelated fungicide-sensitive fungus, Zymoseptoria tritici, doesn’t make sense to me. To point out the possibly unique behaviour of FocTR4, comparison with another Fusarium oxysporum pathovar would have been more informative, as this would have shown whether or not the observations are specific to FocTR4 or a general phenomenon of the species.

Regrding the claim that ‘innate resistance is genetically inherited in a sub-population of "persister" cells’: This claim is neither experimentally nor conceptually sufficiently supported. In the results (Fig. 2m), the authors just show pictures of what they call (Abstract, l. 41) ‚morphologically aberrant hyphae‘, but they don’t document the formation of these hyphae in the presence of fungicides. It is not surprising that hyphae show some degree of abnormal grow in the presence of inhibitors. I don’t see a picture showing the response of normal hyphae (including LIVE/DEAD stainings as shown for spores in Fig. 6) after fungicide treatment to support this claim (l. 240) ‚of a majority of fungicide-sensitive cells and a subpopulation of fungicide-tolerant“persisters“.‘ Do normally growing hyphae die, or they continue to grow as persisters as well?! In this context, the authors suggest that most fungicides are only fungistatic rather than fungitoxic. So, a simple explanation would be that these fungicides inhibit and kill growing hyphae but allow survival of resting structures and formation of slow-growing ‚aberrant‘ hyphae derived from spores. In Fig. 6, the survival of spores in the presence of fungicides is documented; it can be expected that after further incubation, ‚aberrant hyphae‘ would have been formed.

Reviewer #2: This manuscript is a relevant piece of work in plant pathology and addresses concepts of how to control the causative agent of Panama disease of banana. The work combines excellent cell biology with infection assays, and examines the efficacies of a large number of fungicides in detail. The combination of cell biology and transcriptomics explain the insensitivity of Fusarium oxysporum f. sp. cubense Tropical Race 4 (FocTR4) to a broad range of market-leading fungicides, i.e. 12 single-site and 9 multi-site fungicides. Based on their findings, the authors suggest disease control strategies focusing on the application of lipophilic cations.

In addition to the excellence in cell biology, a strength of the manuscript is the extensive study with large numbers of banana plants under defined conditions in the greenhouse, and the experiments employing many different fungicidal compounds. Collectively, a very impressive study.

Reviewer #3: Overall, this paper is an excellent contribution to our understanding of FocTR4 resistance to fungicide. The study reveals many interesting aspects of gene function for fungicide resistance. However, it lacks any functional validation studies to confirm these gene models. Additionally, too many important details are hidden in the supplementary methods. It is difficult to understand the in planta tests, in particular. Be aware that temperature is a factor in phytotoxicity. What is not phytotoxic in cooler laboratory conditions, may be so in typical banana growing weather conditions in countries affected by Fusarium Wilt of Banana.

**Part II – Major Issues: Key Experiments Required for Acceptance**

Reviewer #1: 1. The concept and biogenesis of 'morphologically aberrant hyphae' / 'persister cells', as compared to other hyphae, needs to be experimentally substantiated.

2. Comparison with Zymoseptoria tritici should be removed

3. Testing another common F. oxysporum pathovar for a similar or different fungicide response would be highly desirable.

Reviewer #2: Although several details need the author's attention, this work does not require additional experiments.

Reviewer #3: No major changes.

**Part III – Minor Issues: Editorial and Data Presentation Modifications**

Reviewer #1: l. 31/32: Should read ‘This could be due to insensitivity...’

l. 45: Should read ‘Zymoseptoria tritici...’

Fig. 1f: Explain on what basis the ‘principal component analysis of RNA-preparations’ was made.

Fig. 2c: In the legend, it says ‘The central region show spores’, but doesn’t specifiy what kind of spores. The SEM picture shows hyphae and spore-like branches, however they neither look like microspores nor like macrospores.

Fig. 2f/ Fig. 6: Benzimidazoles seem to be effective against growing hyphae but not resting spores; therefore, they might be still be an option to be used in mixtures with multi-site fungicides.

Fig. 3: Spores are resting structures which have a largely different gene expressiong profile compared to growing hyphae, which makes it little surprising that expression profiles of stressed and unstressed hyphae are much more similar to each other.

Fig. S5/S6: It is difficult to compare gene expression changes in response to fungicide treatments in two different fungi with different fungicide sensitivities. As expected, upregulation is observed in both organisms

L. 148ff: Based on a publication from Ploetz (2015a), that chlamydospores are most relevant for F. oxysporum infection, they performed microscopy of chlamydospore infections using GFP-GocSso1-expresssing cells. Later, they show that infection can also occur with similar efficiency when using micro- or macrospores or hyphae, although they all show a ‘distinctive transcriptional profile’. It would have been consequent to do microscopy also with these spore/ hyphal types to confirm the data with chlamydospores, otherwise I don’t see the point to show these experiments, and to do the RNAseq analysis.

L. 162: ‘To this end, we developed protocols to purify the 3 spore forms and hyphae (~94-100% pure; Fig. 1e)’ I can understand how the authors purified the 3 spore forms; but what about hyphae? The hyphae shown in Fig. 1e are forming a mycelium that cannot be counted and applied in the same ‘numbers’ as the spores. Were the hyphae fragmented before counting and inoculation?

l. 227: The authors asked and tested for the existence of possible mutations in fungicide target genes that would explain the resistance of persister cell. This seemed to be not necessary as the authors have shown that persister cells can return to normal growth of hyphae sensitive to fungicides.

l. 249: The principal component analysis of the RNAseq data should be explained in the results, at least I am not familiar with it.

l. 582 (Discussion): ‘Moreover, we report that the ability to cope with high fungicide load is not genetically inherited‘. This sentence contradicts the conclusions in the Abstract (l..39): ‚innate resistance is genetically inherited in a sub-population of "persister" cells’.

- Fig. 6: The Live stainings of the different spore types in are interesting, but they are missing for ‚normal‘ hyphae after fungicide treatments. Furthermore, assays for germination rates would give a biologically more relevant and quantative result. One might have also considered to test the response of fungicide mixtures.

Reviewer #2: The following points need the authors' attention:

Line 44: The authors mention fungicide-detoxifying enzymes. This term is difficult and needs a clear definition or should be modified (see below).

Line 47: The term pathogenicity is qualitative. In case quantitative differences of aggressiveness are to be compared, the correct term is virulence.

Line 108: The best way to control fungal pathogens …

Line 123-125: This statement does not make a lot of sense. If mutated genes are not expressed, there would be no phenotype. Moreover, not only mutations in target genes are relevant, but also mutations in transcription factors can cause overexpression of fungicide target genes. See the work by Kretschmer et al. (2009), PLoS Pathog. 5:e1000696. This work should be discussed and cited.

Line 138: Here MALCs are highlighted as effective chemistries against FocTR4. In line 48, captan is also recommended. Why is this compound not included here?

Line 150: Replace infected by inoculated.

Lines 152-153: Delete sentence.

Lines 157-158: Without hyphae, how can invasive growth occur? Delete sentence.

Line 173: Spore suspension.

Line 174: … even distribution in the inoculated soil.

Line 187: , and/or a hindering effect of soil: Explain or delete.

Line 204, and below: ED100 values are difficult to determine, due to asymptotic nature of graphs. Therefore ED90 values, in addition to ED50 values, are usually given and should be provided here as well.

Lines 215, Table 1: Give two decimals only.

Lines 236 onwards: To my understanding, certain hyphae activate mechanisms allowing them to grow in the presence of fungicides. They may be called persisters but could also be called fungicide-resistant/tolerant strains or resistant/tolerant hyphae. The alteration of morphology occurs only in the presence of fungicides, correct? And such alterations, as, for example, caused by azole fungicides, are known since long (Kang et al. 2001, Pest. Manage. Sci. 57:491-500), and it is not very surprising that azole applications affect plasma membrane function and, hence, cell wall rigidity. This applies also to the legend of Fig. 3A and should be clearly stated.

Lines 251: In principally all cases I am aware of, fungicide resistance has only been tested in growing hyphae. In dormant structures (distinct spores), fungicide resistance is difficult to address. So, is it really surprising that hyphae, and not the spores tested, have fungicide resistance-related transcript profiles? The authors should discuss this.

Lines 254 and following: Is the term inherited correctly used here? As an alternative, one could say that resistance is not mutation-based. The term inheritance suggests that sexual crosses have been made.

Line 271: … fluxapyroxad or 500 µg …

Lines 323 onward: The authors discuss technical variations. Have RT-qPCR analyses been done to confirm the RNA-Seq data?

Line 340 onward: Here, the term detoxifying enzymes comes into focus. To my feeling, this should be defined more precisely. I fully agree that P450 enzymes belong in this group of detoxifying enzymes. But what about sterol biosynthesis enzymes? And what about transporters? These reduce intracellular drug concentrations, but is the term detoxification correct to describe this? Moreover, a methyltransferase with homology to a polyketide synthase (?) is thought to serve detoxification. The authors may wish to discriminate mechanisms conferring resistance more clearly.

Line 548: Replace the term cell suicide by the term cell death.

Line 594: … this pump possibly participates …

Line 621: … al. (2021) applied …

Line 641: What does the term 'and the soil per se mean? Ion exchange complexity? Microbial complexity? Please define.

Lines 653-656: Not only captan, but principally all pesticides affect non-target organisms. This sentence should be deleted.

Line 1251: … azoxystrobin or 500 µg …

Line1329, Figure 5B: Green and red letters on a grey background are difficult to read.

Reviewer #3: Historically, Panama disease refers specifically to Race 1 and should be referred to instead as Fusarium Wilt of Banana

Line 31 This could be due "to" insensitivity of the pathogen to fungicides and/or soil application per se.

Line 44 Comparison of gene expression in FocTR4 and "the fungicide sensitive" - "Zymoseptora" tritici.....

Line 48, At what concentration was captan and lipophilic cations effective?

Line 64, the use of "unusual hyphae" is vague.

Please consider adding in a better descriptive term for the hyphae.

Line 142: What isolate of FocTR4 is being used and where was it acquired (please state in the methods section)

Line 143 I'd suggest "fluorescent-tagged" version of.. for clarity

Line 177-178 Any reference for image analysis?

Line 179 Replace “marker” with “indication”

Line 190 We included the Fungicide- sus"c"eptible

Table 1 Include a column indicating groupings of fungicides to 7 main classes

Fig 2 Change y-axis of all graphs to maximum 120

Line 237 "with strong inhibition at low concentrations, but sparse growth at higher concentrations" paradoxical growth.

Line 237 Clarify why epoxiconazole not included in the statement but included in the test and succeeding discussion

Line 247 please add additional description for the morphologically aberrant hyphae, such as swollen, deformed, ect.

Figure 3a Please add a control growth image for comparison.

Line 281 Remove “,” before (Steinhauer et al., 2019)

Line 302 Remove “.” after (L84W. Fig. 4b)

Line 305 Replace “.” with “and” between Foc_Erg11/3 and Foc_Sdh3/2.

Line 305 Indicate which gene is for azole and which is for SDHI resistance. Can just add “respectively”

Line 307 Where those untreated control cells hyphae? Please specify

Line 384 up-regulation of Foc_Sdh3/2 subunit strongly suggests SDHI-tolerance. Further molecular characterization is needed for a cause/effect relationship.

Line 430: “Tetramethylrhodamine, methyl ester (TMRM) dye signal” (or specify in the methods)

Line 455 Change color of dot from red to black

Line 471 Please include stat analysis used for Fig 6g

Line 495 Please elaborate on “obvious health differences from plants treated with water”

Table 2: Please include a column reporting differences in transcriptomic levels

For section: “All three FocTR4 447 spore forms survive treatment with most fungicides.” Were any germination tests run to confirm if any of the fungicides blocked germination?

Line 502 Please clarify if day 0 and day 7 is after inoculation

Line 508 There is no Fig 7d

Line 559: see above comment about ‘aberrant hyphae’.

Line 687: Fungal strains (FocTR”4”,... please list the source of the strain used

Line 738 Please include at what temperature was the phytotoxicity tests were performed considering tropical conditions for growing bananas

Line 743 - 745: Please include more detail on how the fungicide was applied, what time frame.

PLOS authors have the option to publish the peer review history of their article (what does this mean?). If published, this will include your full peer review and any attached files.

Reviewer #1: **Yes: **Matthias Hahn

Reviewer #2: No

Reviewer #3: No
---

## [Decision Letter · Decision Letter 1]

30 Aug 2022

Dear Prof. Steinberg,

Thank you very much for submitting your manuscript "Multi-site fungicides suppress banana Panama disease, caused by Fusarium oxysporum f. sp. cubense Tropical Race 4" for consideration at PLOS Pathogens. As with all papers reviewed by the journal, your manuscript was reviewed by members of the editorial board and by two independent reviewers. The reviewers appreciated the attention to an important topic. Based on the reviews, we are likely to accept this manuscript for publication, but would like to give you the opportunity to make a number of modifications based on the minor suggestions of one of the reviewers.

Sincerely,

Bart Thomma

Section Editor

PLOS Pathogens

Bart Thomma

Section Editor

PLOS Pathogens

Kasturi Haldar

Editor-in-Chief

PLOS Pathogens

orcid.org/0000-0001-5065-158X

Michael Malim

Editor-in-Chief

PLOS Pathogens

orcid.org/0000-0002-7699-2064

Reviewer Comments (if any, and for reference):

Reviewer's Responses to Questions

**Part I - Summary**

Reviewer #1: The authors have done major efforts to address the comments of the reviewers, and now present a significantly improved, high quality manuscript that is well-suited for publication in PLoS Pathogens.

Reviewer #2: As before, the study is of significant relevance in both mechanistically understanding fungicide resistance in the highly devastating strain FocTR4 as well as in improving disease control in the field. The strength of this study is the wealth of data and, this may be more important, the high quality of cell biology experiments (microscopy).

**Part II – Major Issues: Key Experiments Required for Acceptance**

Reviewer #1: none

Reviewer #2: I do not see further experiments required in order to accept this manuscript. However, minor issues exist (see below).

**Part III – Minor Issues: Editorial and Data Presentation Modifications**

Reviewer #1: none

Reviewer #2: Comments on the revised manuscript entitled “Multi-site fungicides suppress banana Panama disease, caused by Fusarium oxysporum f. sp. cubense Tropical Race 4” by Stewart Cannon et al.

The revised manuscript by Cannon et al. has significantly improved; yet, some minor details need the attention of the authors.

The only general suggestion is to modify the text with respect to the many times the word “we” is used (six times in the paragraph lines 179-192; seven times on page 7; there are several examples). This I found disturbing as it takes away the focus from the data. The authors may wish to re-consider. The inclusion of ED90 values is appreciated.

In the Introduction, the authors explain what chlamydospores and macro- and microconidia are and how they infect their host plant (Lines 96 – 105). I suggest deleting this section, as plant pathologists should know and not need this information.

Lines 122-123: … related to the mode(s) of action of such fungicides within …

Line 125: … enzyme catalyzing an essential process.

Line 139: “We compared FocTR4 and Z. tritici transcriptional responses”: Why were FocR1 or any other fungicide-sensitive Foc strain not included?

Line 158: 7 days post inoculation.

Line 183: We inoculated …

Line 228 – Table 1: Would it make sense to add standard deviations? The authors give three decimals to indicate the preciseness of the data shown. However, SDs better reflects the preciseness.

Line 241: … most FocTR4 formed hyphae... Do the authors mean: … most FocTR4 microconidia formed hyphae?

Line 324: This suggests that strong …

Line 559: FocR1

Fig. 8b: What does plant health (%) mean? Does it mean %age of plants without disease symptoms? The authors call it a subtle phenotype (line 536), so the term plant health (%) may not be fully appropriate.

PLOS authors have the option to publish the peer review history of their article (what does this mean?). If published, this will include your full peer review and any attached files.

Reviewer #1: **Yes: **Matthias Hahn

Reviewer #2: No

Figure Files:

Data Requirements:

Reproducibility:

References:

---

## [Editor Report · Decision Letter 2]

6 Sep 2022

Dear Prof. Steinberg,

dear Gero,

We are pleased to inform you that your manuscript 'Multi-site fungicides suppress banana Panama disease, caused by Fusarium oxysporum f. sp. cubense Tropical Race 4' has been provisionally accepted for publication in PLOS Pathogens.

Best regards,

Bart Thomma

Section Editor

PLOS Pathogens

Bart Thomma

Section Editor

PLOS Pathogens

Kasturi Haldar

Editor-in-Chief

PLOS Pathogens

orcid.org/0000-0001-5065-158X

Michael Malim

Editor-in-Chief

PLOS Pathogens

orcid.org/0000-0002-7699-2064
---

## [Editor Report · Acceptance letter]

26 Sep 2022

Dear Prof. Steinberg,

We are delighted to inform you that your manuscript, "Multi-site fungicides suppress banana Panama disease, caused by *Fusarium oxysporum* f. sp. *cubense* Tropical Race 4," has been formally accepted for publication in PLOS Pathogens.

Best regards,

Kasturi Haldar

Editor-in-Chief

PLOS Pathogens

orcid.org/0000-0001-5065-158X

Michael Malim

Editor-in-Chief

PLOS Pathogens

orcid.org/0000-0002-7699-2064